# To Tackle Adversarial Transferability: A Novel Ensemble Training Method with Fourier Transformation

**Wanlin Zhang**[1,3]**, Weichen Lin**[2]**, Ruomin Huang**[4]**, Shihong Song**[1]**, Hu Ding**[1]*

[1]School of Computer Science and Technology, University of Science and Technology of China
[2]School of Artificial Intelligence and Data Science, University of Science and Technology of China
[3]Shanghai Innovation Institute    [4]Department of Computer Science, Duke University
`{ideven, linweichen, shihongsong}@mail.ustc.edu.cn`
`ruomin.huang@duke.edu, huding@ustc.edu.cn`

## Abstract

Ensemble methods are commonly used for enhancing robustness in machine learning. However, due to the "transferability" of adversarial examples, the performance of an ensemble model can be seriously affected even it contains a set of independently trained sub-models. To address this issue, we propose an efficient data transformation method based on a cute "weakness allocation" strategy, to diversify non-robust features. Our approach relies on a fine-grained analysis on the relation between non-robust features and adversarial attack directions. Moreover, our approach enjoys several other advantages, e.g., it does not require any communication between sub-models and the construction complexity is also quite low. We conduct a set of experiments to evaluate the performance of our proposed method and compare it with several popular baselines. The results suggest that our approach can achieve significantly improved robust accuracy over most existing ensemble methods, and meanwhile preserve high clean accuracy.

## 1 Introduction

In the past decade, *Deep neural networks (DNNs)* have achieved prominent performance on a broad range of real-world tasks (Goodfellow et al., 2016). However, a number of previous works show that DNNs are susceptible to carefully-crafted manipulations, where the manipulated data are called "adversarial examples" (Szegedy et al., 2014; Zhou et al., 2018; Heaven, 2019). The existence of adversarial examples severely impedes the application of DNNs in security-conscious scenarios, such as self-driving car (Rossolini et al., 2023; Zhu et al., 2021) and heath care (Newaz et al., 2020).

The *adversarial training* approach (Wang et al., 2023a; Madry et al., 2018) has gained significant attention due to its great effectiveness for defending against adversarial examples. However, the adversarial training approach often necessitates considerably high training time and large training dataset (Gowal et al., 2021; Carmon et al., 2019). Moreover, it has been observed that adversarial training is likely to incur certain decline in the accuracy on clean data, which also hinders the trained model to be applied for many practical tasks (Tsipras et al., 2018; Zhang et al., 2019).

Another important approach to enhance adversarial robustness is *ensemble training* (Tramèr et al., 2018). But recent studies (Yang et al., 2025; Gao et al., 2022; Waseda et al., 2023) demonstrated that an adversarial example can attack different models even they are trained independently, and this phenomenon is the so-called "**transferability**" of adversarial examples. Hence, the strategy that simply integrates different models trained on the same original dataset is not sufficient to guarantee the overall robustness. To resolve this issue, different approaches have been proposed for maximizing the "diversity" among sub-models; in general, these approaches can be categorized into two classes: "simultaneous training" and "individual training" (Pang et al., 2019).

To reduce the similarity among sub-models, most existing "simultaneous training" methods attempt to incorporate some penalty during each epoch of parameter updates. Kariyappa & Qureshi (2019)

---

*Corresponding author.

proposed the "Gradient Alignment Loss (GAL)" method to minimize the gradient similarity between sub-models directly. Further, Yang et al. (2021) proposed the "Transferability Reduced Smooth (TRS)" method to improve GAL by adding a regularization term to increase the smoothness, as the models with a smoother loss function can reduce the "transferability" of attacks. Yang et al. (2020) aimed to isolate the adversarial vulnerability in each sub-model by distilling non-robust features, where the sub-models can then generate diverse outputs being resilient against transfer attacks. Despite their effectiveness for defending adversarial attacks, the simultaneous training methods often require a substantial amount of memory since all the sub-models need to be stored in the GPUs in the training stage, which could be prohibitive if the number of sub-models is not small (say, more than 10) and/or their sizes are large. Additionally, the information interaction in parallel training can also cause extra large communication cost.

Different from simultaneous training, most "individual training" methods train each sub-model independently on a randomly transformed version of the given training dataset (Pang et al., 2019; AprilPyone & Kiya, 2021). This "random transformation" strategy yields diverse datasets, and thus different sub-models trained on these datasets can present diverse performances when confronting an adversarial attack. The individual training approach has higher flexibility and also requires less GPU memory, because the sub-models do not need to be stored simultaneously. Since there is no communication between sub-models, individual training methods are more suitable for parallel training with multiple GPUs. But unfortunately, recent studies showed that the commonly used random transformations (e.g. image cropping and rescaling) are not that effective under adversarial attacks (Athalye et al., 2018). The major cause of suppressing the performance of individual training is that the "transferability" problem is still not well addressed.

**Our contributions.** To tackle the transferability obstacle, we consider developing a new data transformation method for ensemble training. Our main contributions are summarized as follows:

– First, we propose a fine-grained analysis on the relation between non-robust features and adversarial attack directions (Section 3). Being different from the previous analysis on non-robust features, our new analysis provides us the hints that are particularly useful to allocate the potential vulnerability directions to a set of sub-models, and therefore paves the way for designing our ensemble training strategy.

– Second, we propose a data transform framework that can effectively promote the diversity of training data for robust ensemble training. The framework consists of two steps: "frequency selection" and "frequency transformation", where the frequency is based on the Fourier transformation on the images. We propose two efficient frequency transformations with low complexities on the identified non-robust features. The first one is based on simple random noise, and the second one is a cute "targeted attack transformation" that can modify the non-robust features more effectively (Section 4.2).

– Finally, we conduct a set of experiments to evaluate the adversarial robustness of our approach on several benchmark datasets under the widely used attack algorithms. We also compare our approach with several open-source ensemble methods, such as ADP (Pang et al., 2019), GAL (Kariyappa & Qureshi, 2019), DVERGE (Yang et al., 2020), and TRS (Yang et al., 2021). Compared with those baselines, the experimental results suggest that our proposed approach can significantly outperform most of them in robust accuracy and also preserve comparable high clean accuracy.

## 1.1 OTHER RELATED WORKS

**Data transformation for ensemble training.** Guo et al. (2018) and Raff et al. (2019) proposed the transformations that preserve semantic information to reduce the impact of adversarial perturbation. AprilPyone & Kiya (2021) developed a training method that employs block-wise data transformations, where the input image is partitioned into blocks based on some private key. LINAC (Rusu et al., 2022) uses a predetermined random seed (private key) to initialize and train a DNN to encode the input data, serving as an encrypted input transformation.

**Adversarial attack from frequency perspective.** Wang et al. (2020) explained that the model's vulnerability to small distortions may be due to its dependence on high-frequency features. Yucel et al. (2023) proposed a data augmentation method that reduces the reliance on high-frequency components, so as to improve model's robustness while maintaining clean accuracy. Maiya et al. (2021) and Bernhard et al. (2021) respectively showed that to fully understand the vulnerability, we should consider the distribution of the entire dataset with high and low frequencies.

## 2 PRELIMINARIES

**Some notations.** We consider the $k$-classification task: $\mathcal{X} \to \mathcal{Y}$ where $\mathcal{X}$ is the input data space and $\mathcal{Y} = \{1, 2, \ldots, k\}$ is the set of labels. A soft-classification model $\boldsymbol{f}(\cdot; \boldsymbol{\beta})$ maps each $x \in \mathcal{X}$ to a vector $f(x; \beta) \in \mathbb{R}^k$, where $\beta$ is the parameter vector that needs to be trained. Its associated hard-classification model is $\boldsymbol{F}(\boldsymbol{x}; \boldsymbol{\beta}) = \arg\max_i [f(x; \beta)]_i$ where $[\cdot]_{\boldsymbol{i}}$ stands for the $i$-th coordinate. The model $f$ is usually equipped with a loss function $\boldsymbol{\ell}(\boldsymbol{f}(\boldsymbol{x}; \boldsymbol{\beta}), \boldsymbol{y})$, $x \in \mathcal{X}$ and $y \in \mathcal{Y}$, which is differentiable on $\beta$ (e.g., cross-entropy loss). We refer to the accuracy on the original dataset as "**clean accuracy**" and the accuracy on adversarial examples as "**robust accuracy**". We denote the one-hot $k$-dimensional vector that corresponds to the target label $y$ as $\boldsymbol{h}(\boldsymbol{y})$.

**Definition 2.1 (Ensemble Model)** *Let* $\mathcal{M} = \{f_1, ..., f_M\}$ *be a set of sub-models for a $k$-classification task. We build the ensemble model with the following function:*

$$f_{\mathrm{E}}(x; \beta_{[1:M]}) = \frac{1}{M} \sum_{m \in [M]} \widehat{F}_m(x; \beta_m),\tag{1}$$

*where* $\beta_{[1:M]} = \{\beta_m \mid 1 \leq m \leq M\}$*, and* $\widehat{F}_m(x; \beta_m)$ *is the one-hot $k$-dimensional vector of the hard-classification model* $F_m(x; \beta_m)$ *of* $f_m$*.*

**Definition 2.2 (Adversarial Attack and Targeted Attack)** *Given a model* $f(\cdot; \beta)$ *and an input* $(x, y) \in \mathcal{X} \times \mathcal{Y}$*, the* ***adversarial attack*** *algorithm* $\mathcal{A}$ *returns a perturbed data* $x'$ *inside the $l_p$ ball of radius* $\epsilon > 0$*, which maximizes the loss function* $\ell(f(\cdot; \beta), \cdot)$*, or minimizes the loss function* $\ell(f(\cdot; \beta), y_t)$ *if given a target label* $y_t \neq y$*. For the latter one, we say it is a "**targeted attack from** $y$ **to** $y_t$". Usually we set $p = 2$ or $p = \infty$.*

As mentioned in Section 1, because our proposed approach is based on Fourier transform, we introduce several necessary notations below. Given an image $x$ of size $L \times N$, the corresponding two-dimensional discrete Fourier transform can be written as: for any $0 \leq u \leq L - 1$ and $0 \leq v \leq N - 1$,

$$\tilde{x}[u, v] = \sum_{s=0}^{L-1} \sum_{t=0}^{N-1} x[s, t] \cdot e^{-2\mathbf{j}\pi\left(\frac{us}{L} + \frac{vt}{N}\right)},\tag{2}$$

where "$\mathbf{j}$" denotes the imaginary unit, and "$\tilde{x}[u, v]$" is the entry in the $u$-th column and $v$-th row of the Fourier matrix $\tilde{x}$ ("$x[s, t]$" is defined similarly for the original image $x$). The pixels of the image $x$ form the **time domain**, and the entries of $\tilde{x}$ form the **frequency domain**. For a frequency $(u, v)$, the **amplitude** is the absolute value $|\tilde{x}[u, v]|$. We call a frequency $(u, v)$ as a **frequency feature**.

## 3 FINE-GRAINED ANALYSIS ON ENSEMBLE MODEL VULNERABILITY

The previous work (Ilyas et al., 2019) categorizes the features learned by a model into robust and non-robust features. It shows that adversarial vulnerability is a natural consequence of the presence of highly predictive but non-robust features. Moreover, different models trained on the same dataset often have similar non-robust features, and therefore an adversarial example usually exhibits the "transferability" property among them. Several other works also presented detailed discussions on the impact of non-robust features (Benz et al., 2021; Springer et al., 2021). Following those studies, a natural idea for tackling the transferability issue is to ensure that the sub-models should have **diverse non-robust features**. In this section, we provide a fine-grained analysis on the vulnerability of ensemble models and then conclude two important hints for achieving this "diversity" goal.

The following definitions are inspired by (Ilyas et al., 2019). Note that different from the term "feature" used in their article, we use "feature extractor" instead in our paper, since "feature" will be particularly used for referring to image feature in time or frequency domain. Specifically, we define a "feature extractor" as a function that maps the input $x \in \mathcal{X}$ to a vector in $\mathbb{R}^k$. A model $f$ is composed of a set of different feature extractors, with each feature extractor focusing on distinct feature. The combination of outputs from these feature extractors forms the model's final output. We then further define the "useful feature extractors".

**Definition 3.1 (Useful feature extractor)** *For a given data distribution* $\mathcal{D} = \mathcal{X} \times \mathcal{Y}$*, a feature extractor* $\theta : \mathcal{X} \to \mathbb{R}^k$ *is* ***useful***, *if we have*

$$\mathbb{E}_{(x,y) \sim \mathcal{D}} \left[ h(y)^\top \theta(x) \right] > \frac{1}{k}.\tag{3}$$

Recall that $h(y)$ is the one-hot $k$-dimensional vector of the label $y$. Roughly speaking, the inequality (3) implies that the expected contribution of a useful feature extractor to the model's correct prediction is higher than the average contribution over all the $k$ classes.

**Definition 3.2 (robust and non-robust feature extractor)** *We use $\mathcal{A}(x)$ to denote the adversarial example of a data item $x$ as described in Definition 2.2. Let $\theta$ be a useful feature extractor. **(1)** We say $\theta$ is **robust** if the following condition holds for any $i$ ($1 \leq i \leq k$):*

$$\mathbb{E}_{(x,y)\sim\mathcal{D}_i}\left[\theta(\mathcal{A}(x))\right]_i > \frac{1}{k}$$

*where $\mathcal{D}_i$ represents the $i$-th class data. We denote the set of these robust feature extractors as $\mathbf{\Theta_R}$.*

*(2) The remaining useful feature extractors are **non-robust**. We assign these non-robust extractors to $k(k-1)$ sets: $\{\mathbf{\Theta_{i,j}} \mid \mathbf{1 \leq i \neq j \leq k}\}$ as follows. Initially, all these $k(k-1)$ sets are empty. Then we go through all the non-robust feature extractors. For each non-robust $\theta$, there must exist at least an index "$i$" such that*

$$\mathbb{E}_{(x,y)\sim\mathcal{D}_i}[\theta(\mathcal{A}(x))]_i \leq 1/k;$$

*we let $j = \arg\max_s \mathbb{E}_{(x,y)\sim\mathcal{D}_i}[\theta(\mathcal{A}(x))]_s$ and assign $\theta$ to $\Theta_{i,j}$ (note that $j$ should be not equal to $i$, or there are multiple indices achieving the maximum expectation and at least one is not equal to $i$, since otherwise $\sum_{s=1}^k \mathbb{E}_{(x,y)\sim\mathcal{D}_i}[\theta(\mathcal{A}(x))]_s$ will less than 1). Eventually, these $k(k-1)$ sets are constructed, where each $\Theta_{i,j}$ contains the feature extractors that are not robust to the attack from $i$ to $j$.*

**Remark 3.3** *Intuitively, if a feature extractor is robust, it should have the capability to preserve its contribution to the correct prediction even under perturbation. It is also worth noting that a non-robust feature extractor $\theta$ could be assigned to multiple $\Theta_{i,j}$s.*

Assume we have a standardly trained model $f$ consisting of a set of useful feature extractors, and we denote it as $\Theta_f$. Each of them can be classified as robust or non-robust as Definition 3.2. Similar with the formulation proposed in (Ilyas et al., 2019), we can represent the model as

$$f(x) = \sum_{\theta\in\Theta_R\cap\Theta_f} w_\theta\theta(x) + \sum_{i,j=1,i\neq j}^{k} \sum_{\theta\in\Theta_{i,j}\cap\Theta_f} w_\theta\theta(x), \tag{4}$$

where each $\theta$ has a coefficient $w_\theta \in \mathbb{R}$. We then conduct our analysis based on Equation (4). Some recent works reveal that adversarial training method can obtain robust model through reducing the dependence on non-robust feature extractors (Allen-Zhu & Li, 2022; Tsipras et al., 2018). However, this strategy may cause certain downgrade performance on clean accuracy (because the non-robust feature extractors also contribute to obtaining correct prediction). Fortunately, we are able to avoid this dilemma in the context of ensemble training. **Namely, we just need to keep the non-robust features as diverse as possible, instead of entirely eliminating the dependence on those non-robust feature extractors.** To pave the way for realizing this goal, we introduce the definition of vulnerability of ensemble model bellow.

**Definition 3.4 (Vulnerability of ensemble model)** *Suppose $f_\mathrm{E}$ is an ensemble model as described in Definition 2.1, and its associated hard-classification model is denoted by $F_\mathrm{E}$: $\forall x$, $F_\mathrm{E}(x) = \arg\max_i[f_\mathrm{E}(x)]_i$. Given the data distribution $\mathcal{D} = \mathcal{X} \times \mathcal{Y}$, **the vulnerability of $F_\mathrm{E}$** is defined as:*

$$\mathrm{Vr}(F_\mathrm{E}) = \mathbb{E}_{(x,y)\sim\mathcal{D}}\Big[\mathbb{I}\big\{F_\mathrm{E}(x) = y \wedge F_\mathrm{E}(\mathcal{A}(x)) \neq y\big\}\Big], \tag{5}$$

*where $\mathbb{I}(\cdot)$ represents the indicator function. Furthermore, for any target class $y_t$, we can define **the vulnerability towards $y_t$** as $\mathrm{Vr}(F_\mathrm{E}, y_t) = \mathbb{E}_{(x,y)\sim\mathcal{D}}\Big[\mathbb{I}\big\{F_\mathrm{E}(x) = y \wedge F_\mathrm{E}(\mathcal{A}(x)) = y_t\big\}\Big]$.*

The vulnerability of Definition 3.4 describes the success probability of an attack $\mathcal{A}$ to the ensemble model $F_\mathrm{E}$. We have the following key inequality, which indicates that $\mathrm{Vr}(F_\mathrm{E})$ is bounded by considering all the attack directions, i.e.,

$$\mathrm{Vr}(F_\mathrm{E}) \leq \sum_{y_t\in\mathcal{Y}} \mathrm{Vr}(F_\mathrm{E}, y_t). \tag{6}$$

The proof of Inequality (6) is placed in Appendix A.1. Moreover, if $F_{\mathrm{E}}(\mathcal{A}(x)) = y_t$, there are at least $M/k$ sub-models returning the wrong label $y_t$ due to the pigeonhole principle. Namely, "$\sum_{m=1}^{M} \mathbb{I}\big([f_m(\mathcal{A}(x))]_y < [f_m(\mathcal{A}(x))]_{y_t}\big) > \frac{M}{k}$" should be a necessary condition for successfully attacking from $y$ to $y_t$. So it implies

$$\mathrm{Vr}(F_{\mathrm{E}}, y_t) \le \mathbb{E}_{(x,y) \sim \mathcal{D}}\Big[\mathbb{I}\Big(\sum_{m=1}^{M} \mathbb{I}\big([f_m(\mathcal{A}(x))]_y < [f_m(\mathcal{A}(x))]_{y_t}\big) > \frac{M}{k}\Big)\Big]. \tag{7}$$

From the upper bound (6), we can decrease the total vulnerability by reducing $\mathrm{Vr}(F_{\mathrm{E}}, y_t)$ for each $y_t$. Also, from (7) we know that $\mathrm{Vr}(F_{\mathrm{E}}, y_t)$ can be reduced by decreasing the chance of "$[f_m(\mathcal{A}(x))]_y < [f_m(\mathcal{A}(x))]_{y_t}$" over $m \in \{1, 2, \cdots, M\}$. According to the Equation (4), the inequality $[f_m(\mathcal{A}(x))]_y < [f_m(\mathcal{A}(x))]_{y_t}$" can be rewritten as

$$\Big[\sum_{\theta \in \Theta_R^m} w_\theta \theta(\mathcal{A}(x)) + \sum_{i,j=1,i\neq j}^{k} \sum_{\theta \in \Theta_{i,j}^m} w_\theta \theta(\mathcal{A}(x))\Big]_y < \Big[\sum_{\theta \in \Theta_R^m} w_\theta \theta(\mathcal{A}(x)) + \sum_{i,j=1,i\neq j}^{k} \sum_{\theta \in \Theta_{i,j}^m} w_\theta \theta(\mathcal{A}(x))\Big]_{y_t}, \tag{8}$$

where $\Theta_R^m$ and $\Theta_{i,j}^m$ respectively denote the sets of robust and non-robust feature extractors for the sub-model $f_m$. Moreover, the set $\Theta_{y,y_t}^m$ should have relatively larger influence to the right-hand side of (8) than other feature extractor set $\Theta_{y,j}^m$ with $j \neq y_t$, due to the outer operator "$[\cdot]_{y_t}$". Therefore, we conclude our first hint as an intuition for reducing $\mathrm{Vr}(F_{\mathrm{E}})$.

***Hint (i):*** *To decrease the vulnerability in the attack direction $y_t$ (i.e., each term $\mathrm{Vr}(F_{\mathrm{E}}, y_t)$ in the upper bound of (6)), it is reasonable to decrease the influence from the non-robust feature extractors of $\Theta_{y,y_t}^m$.*

In Hint (i), a major difference from the previous analysis (Ilyas et al., 2019; Allen-Zhu & Li, 2022) is that, we in particular relate each attack direction $y_t$ to some specific non-robust feature extractors, where the benefit is that these correspondences can effectively help us to build the diverse ensemble model. Moreover, According to the principle of ensemble methods, as long as at least $M/2 + 1$ sub-models are not successfully attacked, the ensemble model will successfully defend against the attack. So we conclude the second hint that is also important for designing our approach.

***Hint (ii):*** *For each attack direction $y_t$, we only need to consider manipulating the training data of $M/2 + 1$ sub-models instead of all the $M$ sub-models.*

Overall, the above Hint (i) & (ii) play the key roles for inspiring our data transformation method in Section 4.

## 4 OUR ENSEMBLE TRAINING METHOD

We first introduce our model and high-level idea in Section 4.1, and then elaborate on the technical details for the data transformations in Section 4.2.

### 4.1 OVERVIEW OF OUR FRAMEWORK

Note that the feature extractors of a model depend on the given training data. Namely, any modification on the features of the training data can implicitly influence the model. Thus, in this section we follow the Hint (i) & (ii) of Section 3 to design an effective data transformation method. The transformation is expected to modify the features of the training data, so as to enhance the robustness of the trained ensemble model. We train a set of distinct sub-models on the transformed training data; these sub-models can be integrated into an ensemble model being robust against adversarial attacks, while preserving the clean accuracy of each sub-model as much as possible. We use "$\pi_m$" to denote the transformation for the $m$-th sub-model, $1 \le m \le M$, and formulate the following problem by slightly modifying Definition 2.1 (replace $x$ by the adversarial example $\mathcal{A}(x)$ for each sub-model):

$$\min \mathbb{E}_{(x,y) \sim \mathcal{X} \times \mathcal{Y}} \ell\Big(\frac{1}{M} \sum_{m \in [M]} \widehat{F}_m(\mathcal{A}(x); \beta_m), y\Big) \tag{9}$$

where $\beta_m$ is obtained by training on the transformed data, i.e., $\beta_m = \underset{\beta}{\mathrm{argmin}} \mathbb{E}_{(x,y) \sim \mathcal{X} \times \mathcal{Y}} \ell\left(f_m(\pi_m(x); \beta), y\right)$ for each $m \in \{1, 2, \cdots, M\}$.

The major challenge for solving the above problem (9) is how to design a set of appropriate transformations $\{\pi_m \mid 1 \leq m \leq M\}$, so that the obtained parameters $\beta_{[1:M]}$ can yield sufficiently diverse sub-models. To address this issue, we leverage the transformation from frequency domain to guide the non-robust features of each sub-model to be as diverse as possible. Specifically, we introduce a method called "**Frequency Domain Transformation (FDT)**" for constructing the set of diverse training datasets $\{\pi_1(\mathcal{X}), \pi_2(\mathcal{X}), \cdots, \pi_M(\mathcal{X})\}$. FDT relies on a key "weakness allocation" strategy. Roughly speaking, the strategy aims to promote the diversity of the constructed datasets, and meanwhile preserve that the overall clean accuracy should not be sacrificed in the ensemble. The details are presented in the next section.

## 4.2 FREQUENCY DOMAIN TRANSFORMATION

Before performing the transformation, we need to select a set of non-robust features. In time domain, a simple observation is that an image feature is usually invariant under spatial translation, e.g., it can appear at different positions in images. This property causes the challenge for directly identifying and representing non-robust features in time domain. Thus we turn our attention to the frequency domain. Moreover, some previous studies on robust learning already revealed that robust and non-robust features are often deeply related to frequency domain (Wang et al., 2020; Bernhard et al., 2021; Maiya et al., 2021).

**Amplitude based selection.** To identify the non-robust frequencies, a straightforward idea is to test the robustness of each individual frequency and select the non-robust ones. Nevertheless, it may takes a large computational cost since the number of frequencies is high (e.g., if the input image is $64 \times 64$, the number of frequencies is also $64 \times 64 \approx 4 \times 10^3$). We propose an easy-to-implement selection idea based on the amplitudes, since the amplitudes can be directly obtained via the Fourier transformation with low complexity. According to the previous research (Ilyas et al., 2019; Benz et al., 2021; Springer et al., 2021), a feature can be regarded as "robust" if it cannot be easily manipulated by small perturbations. We observe that high-amplitude frequency features usually dominate the ground truth of an image. Figure 4 in our Appendix illustrates an example to show that, if we keep high-amplitude frequencies and remove low-amplitude ones, the image is changed slightly even with adding certain noise (i.e., we can still recognize the ground truth from the modified image). This observation suggests that high-amplitude frequency features are more strongly related to the semantic information of image. So in our following approach, we maintain high-amplitude frequency features as "robust features", and select the frequencies with low amplitudes (by setting a threshold "$\tau$") to transform. Moreover, we can conveniently observe the performance changing through varying the threshold $\tau$ in our experiment. Figure1 illustrates the amplitude-based selection.

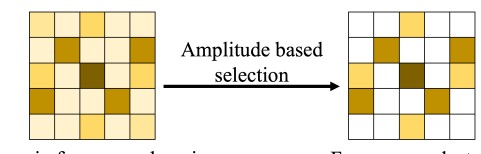

Image in frequency domain          Frequency selected

Figure 1: We use a 5x5 image as a toy example, where the intensity of the color indicates the magnitude of the amplitude. In our amplitude-based selection, we retain the high-amplitude frequencies (i.e., the darker regions) and perform data transformations on the low-amplitude frequencies (i.e., the white regions).

Following our frequency selection, we propose two transformation methods for promoting the diversity by using the identified non-robust features. Our first approach is from a straightforward idea, which is just to replace the non-robust features by random noise (due to the space limit, we leave the details to Appendix C). This method is very easy to implement in practice. Though it can achieve certain degree of improvement upon previous ensemble training methods, the performance is still not very promising (as shown in our experiments). To further improve the effectiveness, we propose a more sophisticated approach called "targeted-attack transformation", which constructs a set of different "substitute" features through attacking the images to different targeted classes, and then use them to replace the selected non-robust frequencies.

**Targeted-attack transformation:** We briefly explain our intuition first. It was shown that adversarial attacks have the capability to manipulate non-robust features (Ilyas et al., 2019; Yang et al., 2020). In particular, a targeted attack as introduced in Definition 2.2, aims at modifying non-robust features that are associated with a specific target label. For instance, let us consider a data point $(x, y)$ in the original dataset $\mathcal{X} \times \mathcal{Y}$; we set the target label as $y_t$ and obtain the corresponding adversarial example

$x'$ ($x'$ contains the modified non-robust features that are associated with $y_t$). When training a model using $(x', y)$, intuitively it can be viewed as an "immunization" for defending the attack from $y$ to $y_t$; and consequently, the chance that obtaining the wrong label $y_t$ for the data with label $y$ decreases. In other words, it becomes more difficult to attack the images with label $y$ to $y_t$ than to the other classes. We call the modified non-robust feature as a "substitute" feature derived by the targeted attack.

Motivated by this observation, we can construct different transformations by using $k \times (k - 1)$ targeted attacks (since each label can be attacked to be the other $k - 1$ labels); these attacks can yield different substitute features, and then we use these features to replace the corresponding non-robust features in the original dataset (based on Hint (i) in Section 3); finally, the $M$ transformed datasets are obtained via an allocation algorithm, where each substitute feature is captured by at least $M/2 + 1$ datasets (based on Hint (ii) in Section 3). Overall, due to the completeness of the $k \times (k - 1)$ targeted attacks, the $M$ sub-models trained on those datasets can guarantee the robustness of the final ensemble solution. We introduce some definitions for our transformation first.

**Definition 4.1 (Strengthen a dataset)** *Let $y_1 \neq y_2 \in \mathcal{Y}$. If a given training dataset $P$ contains at least one adversarial example who has the original label $y_1$ but is misclassified as $y_2$, we say that $P$ has been **strengthened by the attack direction from** $y_1$ **to** $y_2$ ("$\overrightarrow{y_1 y_2}$-direction" for short).*

In other words, if $P$ is not strengthened in $\overrightarrow{y_1 y_2}$-direction, the model trained on $P$ is more likely to be fragile to the targeted attacks from $y_1$ to $y_2$. Also, the dataset $P$ may have not been strengthened in multiple different directions. So we define its "weakness set" $\mathcal{W} = \{\overrightarrow{y_1 y_2} \mid 1 \leq y_1, y_2 \leq k, y_1 \neq y_2$, and $P$ has not been strengthened in $\overrightarrow{y_1 y_2}$-direction$\}$.

**Definition 4.2 (Diversity of weakness sets)** *Given $M$ datasets $\{P_1, P_2 \cdots, P_M\}$ with their corresponding weakness sets $\{\mathcal{W}_1, \mathcal{W}_2, \cdots, \mathcal{W}_M\}$, we define their diversity:*

$$\boldsymbol{Div}(P_1, P_2, \cdots, P_M) = 1 - \frac{|\mathcal{W}_1 \cap \mathcal{W}_2 \cap \cdots \cap \mathcal{W}_M|}{\max\{|\mathcal{W}_1|, |\mathcal{W}_2|, \cdots, |\mathcal{W}_M|\}}.$$

It is easy to see that the higher the value $\boldsymbol{Div}(P_1, P_2, \cdots, P_M)$, the more diverse the corresponding weakness sets. A higher diversity suggests that the vulnerabilities of the $M$ sub-models trained on those datasets are more likely to be different. To achieve a nice performance in terms of both accuracy and robustness, we need to take account of the diversity function "$\boldsymbol{Div}$" for designing the transformations. The basic principle is:

*On the one hand, our transformed datasets should have a sufficiently large number of diverse substitute features, so that one adversarial attack cannot easily capture more than half of the $M$ sub-models. On the other hand, the datasets should also maintain the major information of the original input as much as possible, since otherwise the clean accuracy may decline due to the added substitute features.*

To provide an appropriate trade-off, we propose the following constrained optimization objective: let $\mathbb{C}$ be the set of all the $\binom{M}{\lceil M/2 \rceil}$ combinations of $\lceil M/2 \rceil$-size subsets from $\{1, 2, \cdots, M\}$, and then

$$\max_{P_1, P_2, \cdots, P_M} \quad \min\{|\mathcal{W}_1|, |\mathcal{W}_2|, \cdots, |\mathcal{W}_M|\} \tag{10}$$

$$\text{s.t.} \ \forall \{i_1, i_2, \cdots, i_{\lceil M/2 \rceil}\} \in \mathbb{C}, \quad \boldsymbol{Div}(P_{i_1}, P_{i_2}, \cdots, P_{i_{\lceil M/2 \rceil}}) = 1. \tag{11}$$

We maximize the objective function of (10) because we want to minimize the modification degree for each transformed dataset. Intuitively, a large weakness set indicates that the corresponding dataset is not changed significantly by the transformation, and thus the clean accuracy is likely to be well preserved. The constraint (11) guarantees that any $\lceil M/2 \rceil$ datasets have the intersection $\mathcal{W}_{i_1} \cap \mathcal{W}_{i_2} \cap \cdots \cap \mathcal{W}_{i_{\lceil M/2 \rceil}} = \emptyset$, that is, they do not share any common direction. Consequently, the ensemble solution should be robust to any attack direction. To achieve this twofold goal, we design an efficient allocation strategy together with an attack-guided transformation on the training data. Specifically, the procedure consists of the following two stages.

**Stage (1): allocating the weakness sets to the sub-models.** For each $\overrightarrow{y_1 y_2}$−direction, $1 \leq y_1, y_2 \leq k$, there are at most $\lceil \frac{M}{2} \rceil - 1$ sets that contain this direction (due to the constraint (11)), so the sum $\sum_{1 \leq i \leq M} |\mathcal{W}_i|$ is no larger than $k(k - 1) * (\lceil \frac{M}{2} \rceil - 1)$. Therefore, the maximum value of Eq (10) is no larger than

$$k(k-1) * (\lceil \frac{M}{2} \rceil - 1)/M \qquad (12)$$

based on the pigeonhole principle. We assign the total $k(k-1) * (\lceil \frac{M}{2} \rceil - 1)$ directions (each direction is duplicated to be $\lceil \frac{M}{2} \rceil - 1$ copies) to $M$ sets in a round-robin way, where the number of directions assigned to each set is no larger than the upper bound (12). Please refer to Figure 2 for an example.

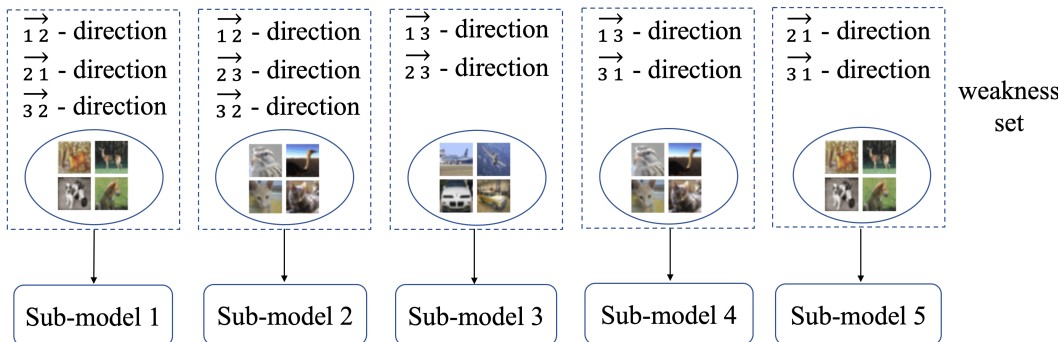

Figure 2: Assign the attack directions to five sub-models for a three-class classification task.

**Stage (2): constructing the new datasets.** Following the allocation, we transform the original dataset, denoted by $P_{\text{ori}}$, to align with the assigned weakness sets for the $M$ sub-models correspondingly. Using the same notations of Definition 4.2, we denote the waiting-to-construct dataset for the $m$-th sub-model as $P_m$ (which is initialized to be $\emptyset$), $1 \le m \le M$. First, we divide $P_{\text{ori}}$ into $k$ subsets $C_1, C_2, \cdots, C_k$, where each $C_j$ corresponds to the label $j$, for $1 \le j \le k$; further, each $C_j$ is equally partitioned to $k-1$ disjoint parts $\{C_{j,1}, C_{j,2}, \cdots, C_{j,k-1}\}$ at random. For each data $(x, j)$ in $C_{j,i}$, we attack it from $j$ to $h$ (let $h = i + j \mod k$) to obtain the adversarial perturbation; then we only substitute the low-amplitude frequencies of $x$ with the perturbation, and other frequencies (which have their amplitudes higher than the aforementioned threshold $\tau$) remain unchanged. We denote the new dataset as $C'_{j,i}$. Finally, we add $C'_{j,i}$ to $P_m$ if the $\overrightarrow{ih}$-direction is not in the weakness set $\mathcal{W}_m$. From the construction method of the weakness sets, we know that the $\overrightarrow{ih}$-direction can appear in at most $\lceil \frac{M}{2} \rceil - 1$ weakness sets. So, the set $C'_{i,j}$ can be added to at least $\lceil \frac{M}{2} \rceil$ different $P_m$s. Consequently, the completeness for defending the $k(k-1)$ attack directions can be guaranteed, i.e., the constraint (11) is satisfied. Figure 3 shows the schematic diagram of the construction process, and the full details are shown in Appendix D.

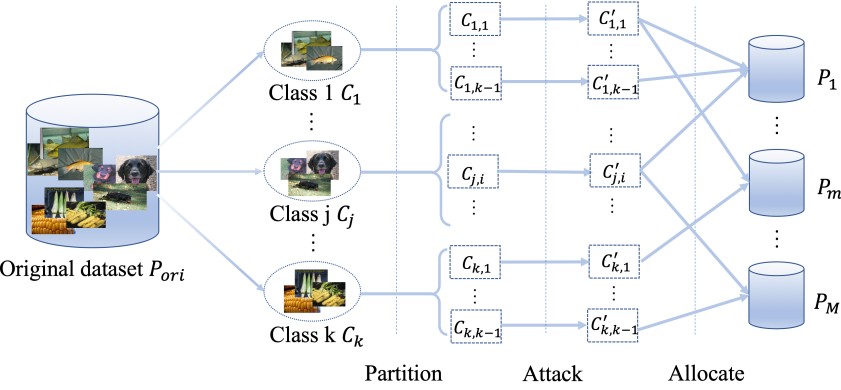

Figure 3: A schematic diagram of the construction process. In the allocation stage, each $C'_{j,i}$ is added to $P_m$ if the $\overrightarrow{ih}$-direction is not in the weakness set $\mathcal{W}_m$, $h = i + j \mod k$.

**Remark 4.3** *We are aware of some previous robust learning approaches that also depend on data modification (Allen-Zhu & Li, 2022; Tsipras et al., 2018). But their approaches usually tend to*

*completely eliminate non-robust features. Our method is quite different, where the goal is to leverage the carefully selected non-robust features to weaken the transferability among sub-models. For each sub-model, we only modify the non-robust features corresponding to certain directions, rather than all non-robust features, and therefore the modification yields relatively lower impact on clean accuracy. Moreover, we partition each class $C_j$ into $k-1$ subsets $\{C_{j,1}, C_{j,2}, \cdots, C_{j,k-1}\}$, with each subset being attacked to a specified class. This step eliminates the need to attack each data point across all classes, thereby reducing the computational complexity of constructing the new datasets.*

## 5 EXPERIMENTS

We conduct our experiments on the widely used image datasets CIFAR-10, CIFAR-100 (Krizhevsky & Hinton, 2009), and Tiny-ImageNet-200 (Deng et al., 2009). As for the baselines, we reproduce the existing ensemble models including ADP (Pang et al., 2019), GAL (Kariyappa & Qureshi, 2019), DVERGE (Yang et al., 2020), and TRS (Yang et al., 2021), with their released codes and recommended hyperparameter settings. As for our approach, "FDT-random" and "FDT-target" respectively denote the methods utilizing random noise based transformation and target-attack transformation; "FDT-hybrid" represents the method that combines both, that is, we set two frequency selection thresholds $\tau_1$ and $\tau_2$ ($\tau_1 < \tau_2$), and perform random and target-attack transformations on the frequencies less than $\tau_1$ and the frequencies between $\tau_1$ and $\tau_2$, respectively (due to the space limit, more details are shown in Appendix E). Our code will be available at https://github.com/ideven123/FDT.

We train each sub-model based on ResNet-20 (He et al., 2016) and use Adam optimizer (Kingma & Ba, 2015) with an initial learning rate of 0.001 for 200 epochs. To further test their performance on neural network with larger scale, we also use WideResNet28-10 (Zagoruyko & Komodakis, 2016) to train the sub-models and the results are placed in our supplement. All the experiments are implemented with PyTorch (Paszke et al., 2017) on a single NVIDIA GeForce RTX 3090 with 24GB of memory and 1TB of storage. We assess the performance of our models through 5 repeated runs and compute error bars. Utilizing the numpy library, we calculate the standard deviation and subsequently derive the standard error of the mean (SEM).

**Varying the number of sub-models.** We take the ResNet-20 model trained on CIFAR-10 as an example and test the performance of FDT with different numbers of sub-models in the ensemble. In this experiment, we set the frequency selection threshold $\tau_1$ to be $0.2$ and $\tau_2$ to be $0.8$. Then we evaluate the performance of FDT-hybrid under FGSM (Madry et al., 2018) , PGD (Carlini et al., 2019), and AutoAttack (AA) (Croce & Hein, 2020) attack methods with $l_\infty$ perturbations of size $\epsilon = 0.02$. The results in Table 1 indicate that our clean accuracy has relatively smaller change as the number increases, while the robust accuracy can be substantially improved from 3 to 20 sub-models.

Table 1: Performance of FDT-hybrid with different sub-model numbers on CIFAR-10.

| Sub-model numbers | 3 | 5 | 8 | 12 | 20 |
|---|---|---|---|---|---|
| Clean accuracy | $90.20 \pm 0.03$ | $90.75 \pm 0.03$ | $91.35 \pm 0.05$ | $91.51 \pm 0.06$ | $91.86 \pm 0.07$ |
| FGSM ($\epsilon = 0.02$) | $58.04 \pm 0.13$ | $61.66 \pm 0.15$ | $62.41 \pm 0.11$ | $63.96 \pm 0.12$ | $64.27 \pm 0.14$ |
| PGD ($\epsilon = 0.02$) | $20.01 \pm 0.04$ | $26.10 \pm 0.07$ | $29.20 \pm 0.05$ | $29.78 \pm 0.08$ | $29.71 \pm 0.07$ |
| AutoAttack ($\epsilon = 0.02$) | $19.42 \pm 0.04$ | $25.37 \pm 0.05$ | $27.33 \pm 0.04$ | $28.12 \pm 0.07$ | $28.92 \pm 0.07$ |

**Results for white-box attack.** To maintain consistency with the baseline ensemble methods from the literature, we ensemble three ResNet-20 sub-models here and evaluate the robust accuracy using $\epsilon = 0.01$ and $\epsilon = 0.02$. In this experiment, we set the frequency selection threshold $\tau_1$ to be $0.2$ and $\tau_2$ to be $0.8$. In the white-box attack setting, the attacker has full knowledge of the models, including model parameters, architecture, and ensemble training strategy. To evaluate the adversarial robustness of the ensemble, we conduct the following white-box attacks: PGD, FGSM, BIM (Goodfellow et al., 2015), MIM (Dong et al., 2018), C&W (Carlini & Wagner, 2017) and AutoAttack (AA). The attacks are implemented using AdverTorch (Ding et al., 2019). We take the robust and clean accuracies, and average training time per epoch as the evaluation metrics.

Table 2 presents the obtained robust accuracies of the baseline ensemble methods on CIFAR-10 and CIFAR-100. In addition, we show the average training time per epoch of different ensemble methods. The experimental results suggest that our FDT-random method can achieve higher adversarial robustness over other baselines on both CIFAR-10 and CIFAR-100, with the training time only higher than ADP (and much lower than other baselines). Furthermore, the FDT-hybrid ensemble method

Table 2: Robust and Clean Accuracy (%) and average training time of different ensemble methods against white-box attacks on CIFAR-10 and CIFAR-100. "$\epsilon$" and "$\lambda$" stand for the $l_\infty$ norm of the adversarial perturbation and the coefficient of C&W attack respectively. The TRS results are reported in the original paper Yang et al. (2021), with "-" indicating results not provided.

| **CIFAR-10** | ADP | GAL | DVERGE | TRS | FDT-random | FDT-target | FDT-hybrid |
|---|---|---|---|---|---|---|---|
| Clean accuracy | **91.84** | 91.81 | 91.37 | - | 89.88± 0.02 | 90.16 ± 0.04 | 90.20± 0.03 |
| FGSM ($\epsilon$=0.01) | 59.48 | 44.97 | 70.05 | - | 66.96± 0.12 | **72.88**± 0.12 | 72.24± 0.12 |
| FGSM ($\epsilon$=0.02) | 53.38 | 30.58 | 56.33 | 44.2 | 46.28 ± 0.10 | 55.54 ± 0.09 | **58.04** ± 0.13 |
| PGD ($\epsilon$= 0.01) | 14.45 | 1.35 | 40.55 | **50.5** | 45.42 ± 0.09 | 46.58 ± 0.07 | 48.48 ± 0.09 |
| PGD ($\epsilon$ = 0.02) | 2.95 | 0.34 | 11.49 | 15.1 | 12.24 ± 0.03 | 15.08 ± 0.05 | **20.01** ± 0.04 |
| BIM ($\epsilon$= 0.01) | 14.15 | 1.37 | 40.51 | **50.6** | 45.24 ± 0.03 | 46.86 ± 0.04 | 48.57± 0.05 |
| BIM ($\epsilon$ = 0.02) | 3.01 | 0.27 | 10.65 | 15.8 | 11.68± 0.03 | 14.86± 0.03 | **16.63** ± 0.02 |
| MIM ($\epsilon$= 0.01) | 20.38 | 2.05 | 44.74 | 51.5 | 47.73± 0.05 | 49.97 ± 0.06 | **51.50** ± 0.07 |
| MIM ($\epsilon$= 0.02) | 5.11 | 0.69 | 14.76 | 17.2 | 15.14 ± 0.04 | 18.27± 0.02 | **20.09** ± 0.03 |
| AA ($\epsilon$= 0.01) | 1.80 | 0.00 | 43.34 | - | 46.09± 0.09 | 48.83± 0.08 | **51.56** ± 0.08 |
| AA ($\epsilon$= 0.02) | 0.00 | 0.00 | 13.72 | - | 9.38 ± 0.05 | 15.70 ± 0.05 | **19.42** ± 0.04 |
| C&W ($\lambda$ = 0.1) | 20.96 | 31.57 | 52.35 | **58.1** | 45.01± 0.10 | 55.48 ± 0.10 | 56.08 ± 0.11 |

| **CIFAR-100** | ADP | GAL | DVERGE | TRS | FDT-random | FDT-target | FDT-hybrid |
|---|---|---|---|---|---|---|---|
| Clean accuracy | 67.04 | **67.70** | 66.16 | - | 66.29 ± 0.11 | 67.64 ± 0.08 | 66.70± 0.09 |
| FGSM ($\epsilon$ =0.01) | 17.82 | 16.89 | 33.94 | - | 35.42± 0.12 | **40.46**± 0.14 | 39.85 ± 0.14 |
| FGSM ($\epsilon$ = 0.02) | 10.53 | 7.80 | 26.61 | 19.3 | 22.40± 0.05 | **32.30**± 0.06 | 30.27± 0.08 |
| PGD ($\epsilon$ = 0.01) | 0.80 | 0.11 | 14.62 | 23.0 | 21.54 ± 0.06 | 22.19 ± 0.05 | **24.93**± 0.05 |
| PGD ($\epsilon$ = 0.02) | 0.01 | 0.02 | 4.25 | 5.3 | 4.84 ± 0.03 | 7.27 ± 0.02 | **8.63**± 0.03 |
| BIM ($\epsilon$ = 0.01) | 0.68 | 0.23 | 14.85 | 22.9 | 21.10 ± 0.09 | 22.39 ± 0.07 | **24.35**± 0.06 |
| BIM ($\epsilon$ = 0.02) | 0.02 | 0.0 | 4.07 | 5.4 | 4.80 ± 0.04 | 6.80± 0.05 | **8.40**± 0.05 |
| MIM ($\epsilon$ = 0.01) | 0.78 | 0.12 | 16.82 | 23.4 | 23.14 ± 0.06 | 24.68± 0.10 | **27.09** ± 0.09 |
| MIM ($\epsilon$ = 0.02) | 0.01 | 0.02 | 5.31 | 6.2 | 6.47 ± 0.03 | 8.87 ± 0.04 | **10.19** ± 0.05 |
| AA ($\epsilon$= 0.01) | 0.01 | 0.00 | 11.23 | - | 16.02± 0.09 | 16.03± 0.09 | **16.41** ± 0.12 |
| AA ($\epsilon$= 0.02) | 0.00 | 0.00 | 2.72 | - | 3.12± 0.04 | 4.54 ± 0.05 | **5.47**± 0.07 |
| C&W ($\lambda$ = 0.1) | 0.74 | 3.70 | 10.68 | 26.9 | 25.07 ± 0.10 | 29.43 ± 0.09 | **30.66**± 0.13 |

| **Time (s)** | ADP | GAL | DVERGE | TRS | FDT-random | FDT-target | FDT-hybrid |
|---|---|---|---|---|---|---|---|
| CIFAR-10 | 30.15 | 69.92 | 134.33 | 350.42 | 37.04 | 108.22 | 114.23 |
| CIFAR-100 | 30.34 | 69.71 | 129.25 | 344.92 | 37.12 | 108.43 | 113.87 |

achieves an even better robustness than FDT-random, though its running time is higher since it needs to perform the target-attack transformation.

**Summary on other experimental results placed in Appendix F.** We also conduct the experiments to examine the performance of FDT under black-box attack, and assess the transferability of our method across various sub-models. The results indicate the competitive robustness of our method in defending against black-box attacks. Then, we evaluate the trade-off between clean accuracy and robust accuracy by varying the frequency selection threshold $\tau$. The result shows that the ensemble model has lower clean accuracy and higher robust accuracy with the increasing of $\tau$. Moreover, we included some ablation studies on datasets and model architectures. These experiments demonstrate that our method performs the best among ensemble-based baseline methods.

## 6 CONCLUSION AND FUTURE WORK

In this paper, we present a novel data transformation approach to improve the robustness of ensemble models against adversarial attacks. By leveraging the frequency based features and strategically allocating adversarial examples, we demonstrate the effectiveness of our method in enhancing adversarial robustness while maintaining high accuracy on clean data. As for the future work, we can consider other types of transformation methods (e.g., beyond using frequency) to improve the ensemble robustness. Also, it is interesting to consider more complicated scenarios for ensemble training, such as federated learning with concerning the privacy issue.

ACKNOWLEDGMENTS

The authors would like to thank the reviewers for their constructive comments and suggestions. This work was partially supported by the National Natural Science Foundation of China (No. 62272432 and No. 62432016), the National Key Research and Development Program of China (No. 2021YFA1000900), and the Natural Science Foundation of Anhui Province (No. 2208085MF163).

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

# A OMITTED PROOFS

## A.1 PROOF OF INEQUALITY (6):

$\sum_{y_t \in \mathcal{Y}} \mathrm{Vr}(F_\mathrm{E}, y_t)$

$= \sum_{y_t \in \mathcal{Y}} \mathbb{E}_{(x,y) \sim \mathcal{D}} \Big[ \mathbb{I}\big\{ F_\mathrm{E}(x) = y \wedge F_\mathrm{E}(\mathcal{A}(x)) = y_t \big\} \Big]$

$= \sum_{y_t \in \mathcal{Y}} \sum_{(x,y) \in \mathcal{D}} p_{(x,y)} \Big[ \mathbb{I}\big\{ F_\mathrm{E}(x) = y \wedge F_\mathrm{E}(\mathcal{A}(x)) = y_t \big\} \Big].$

Then, we interchange the order of summation, and so the above equation is equal to

$\sum_{(x,y) \in \mathcal{D}} p_{(x,y)} \sum_{y_t \in \mathcal{Y}} \Big[ \mathbb{I}\big\{ F_\mathrm{E}(x) = y \wedge F_\mathrm{E}(\mathcal{A}(x)) = y_t \big\} \Big]$

$= \mathbb{E}_{(x,y) \sim \mathcal{D}} \Big[ \sum_{y_t \in \mathcal{Y}} \mathbb{I}\big\{ F_\mathrm{E}(x) = y \wedge F_\mathrm{E}(\mathcal{A}(x)) = y_t \big\} \Big].$

For each $(x, y)$, without loss of generality, let $F_\mathrm{E}(\mathcal{A}(x)) = y_0$. For $y_t \neq y_0$, $\mathbb{I}\big\{ F_\mathrm{E}(x) = y \wedge F_\mathrm{E}(\mathcal{A}(x)) = y_t \big\} = 0$. For $y_t = y_0$, $\mathbb{I}\big\{ F_\mathrm{E}(x) = y \wedge F_\mathrm{E}(\mathcal{A}(x)) = y_t \big\} = \mathbb{I}\big\{ F_\mathrm{E}(x) = y \big\}$. So the above equation is equal to

$\mathbb{E}_{(x,y) \sim \mathcal{D}} \Big[ \mathbb{I}\big\{ F_\mathrm{E}(x) = y \big\} \Big]$

$= \mathbb{E}_{(x,y) \sim \mathcal{D}} \Big[ \mathbb{I}\big\{ F_\mathrm{E}(x) = y \wedge \big( F_\mathrm{E}(\mathcal{A}(x)) \neq y \vee F_\mathrm{E}(\mathcal{A}(x)) = y \big) \big\} \Big]$

$= \mathbb{E}_{(x,y) \sim \mathcal{D}} \Big[ \mathbb{I}\big\{ F_\mathrm{E}(x) = y \wedge F_\mathrm{E}(\mathcal{A}(x)) \neq y \big\} + \mathbb{I}\big\{ F_\mathrm{E}(x) = y \wedge F_\mathrm{E}(\mathcal{A}(x)) = y \big\} \Big].$

We split $\mathbb{I}(.)$ because $F_\mathrm{E}(\mathcal{A}(x)) \neq y$ and $F_\mathrm{E}(\mathcal{A}(x)) = y$ are mutually exclusive. Then, the above equation is equal to

$\mathrm{Vr}(F_\mathrm{E}) + \mathbb{E}_{(x,y) \sim \mathcal{D}} \Big[ \mathbb{I}\big\{ F_\mathrm{E}(x) = y \wedge F_\mathrm{E}(\mathcal{A}(x)) = y \big\} \Big]$

$\geq \mathrm{Vr}(F_\mathrm{E})$

Overall, we can obtain the inequality (6): $\sum_{y_t \in \mathcal{Y}} \mathrm{Vr}(F_\mathrm{E}, y_t) \geq \mathrm{Vr}(F_\mathrm{E})$

# B FREQUENCY SELECTION

Figure 4 illustrates an example to show that, if we keep high-amplitude frequencies and remove low-amplitude ones, the image is changed slightly even with adding certain noise (i.e., we can still recognize the ground truth from the modified image). On the other hand, if we keep the low-amplitude frequencies only, the semantic information is almost missing. This observation suggests that high-amplitude frequency features are more strongly related to the semantic information of image.

# C RANDOM NOISE BASED TRANSFORMATION

**Random noise based transformation:** This approach substitutes the identified non-robust frequencies with Gaussian noise. For an $N \times N$ image, we take the non-robust frequencies based on the pre-specified threshold $\tau$, and replace them with random vector for each sub-model in our experiment. In particular, to further increase the randomness, we perform this transformation for each epoch in the training stage. If we select the top $s$ non-robust frequencies, the overall dimensionality of the edited random feature should be $s \times E$ (we concatenate those $s$-dimensional features together), where $E$ is the number of epochs. For example, if $N = 32$, $s = N^2/2$, and $E = 200$, the overall dimensionality can be as large as $10^5$. Because these $M$ features are random and have high dimensions, they are very likely to be nearly orthogonal with each other (this phenomenon in high-dimensional geometry can be proved by the central limit theorem (Wegner, 2021)). As a consequence, they tend to yield diverse training results for the sub-models.

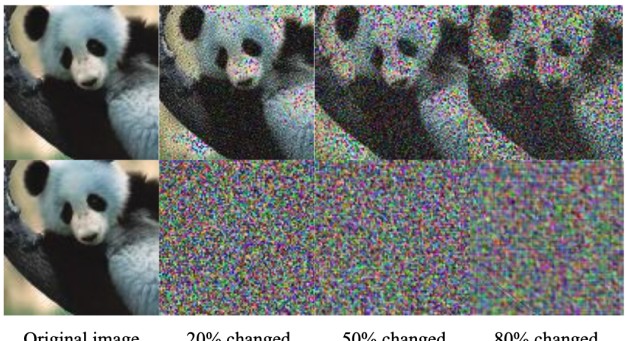

Original image    20% changed    50% changed    80% changed

Figure 4: The first and second rows are the figures by adding random noise to high-amplitude and low-amplitude frequencies, respectively. "20% changed" for the first row means we remove the 20% lowest-amplitudes frequencies, and add small noise to the remaining high-amplitude frequencies. "20% changed" for the second row means we remove the 20% highest-amplitude frequencies, and add small noise to the remaining low-amplitude frequencies. "50% changed" and "80% changed" follow the same procedure as "20% changed".

The implementation details are as follows. Giver an image $x$, we perform Fourier Transform on $x$ and also on a generated Gaussian noise $n_0$. Then, we can obtain the low-amplitude frequencies and high-amplitude frequencies of $x$ by setting an amplitude threshold. Next, we generate two masks ($M_1$ and $M_2$) to select high-amplitude frequencies and low-amplitude frequencies. We add the low-amplitude frequencies of $n_0$ (i.e., $M_2(n_0)$) to high-amplitude frequencies of $x$ (i.e., $M_1(x)$), and obtain the transformation of $x$ (denoted as $\pi(x)$). Finally, we transform $\pi(x)$ to time domain by inverse Fourier transform and train the model with $\pi(x)$.

## D    ALGORITHM OF FDT

Algorithm 1 shows the overall framework of training an ensemble model with FDT. It illustrates that our data transformation is performed at each iteration.

---
**Algorithm 1 Training ensemble model with FDT**

    **Input:** dataset $\mathcal{X} \times \mathcal{Y}$, the number of sub-models $M$, and the epoch number $E$,
    **Output:** sub-model $\beta_1, \beta_2, \cdots, \beta_M$
    **for** $i = 1$ to $E$ **do**
        Run *Targeted-attack Transformation* and obtain $P_1, P_2, \cdots, P_M$ ;
        **for** $j = 1$ to $M$ **do**
            train $\beta_j$ on $P_j$
        **end for**
    **end for**

---

Algorithm 2 shows the details of targeted-attack transformation method on the whole dataset. For each specific image $x$, we obtain the targeted class according to the allocation scheme mentioned in "Stage (1)". Then, we use targeted PGD attack to obtain the adversarial sample $x'$. After that, we perform Fourier Transform on $x$ and $x'$, and we can obtain the low-amplitude frequencies and high-amplitude frequencies of $x$ by setting an amplitude threshold. Next, we generate two masks ($M_1$ and $M_2$) to select high-amplitude frequencies and low-amplitude frequencies. We add the low-amplitude frequencies of $x'$ (i.e., $M_2(x')$) to high-amplitude frequencies of $x$ (i.e., $M_1(x)$), and obtain the transformation of $x$ (denoted as $\pi(x)$). Finally, we transform $\pi(x)$ to time domain by inverse Fourier transform.

---

**Algorithm 2 Targeted-attack Transformation**

---

**Input:** dataset $P_{ori}$, number $M$, steps $s$, class number $k$
**Output:** Transformed data $P_1, P_2, \cdots, P_M$
Divide the dataset $P_{ori}$ into $k$ parts $\{C_1, C_2, \cdots, C_k\}$ according to labels
Randomly partition the dataset $C_j$ equally into disjoint $k-1$ parts $\{C_{j,1}, C_{j,2}, \cdots, C_{j,k-1}\}$
Initialize $P_1, P_2, \cdots, P_M$ with empty set;
$m \leftarrow 0$
**for** $j = 1$ to $k$ **do**
    **for** $i = 1$ to $k-1$ **do**
        $C'_{j,i} \leftarrow$ calculate targeted attack example in $C_{j,i}$ with label $i + j \mod k$ and perform data transformation on each image;
        **for** $s = 1$ to $\lceil \frac{M}{2} \rceil + 1$ **do**
            $m \leftarrow m + 1 \mod M$;
            Append $C'_{j,i}$ to $P_m$;
        **end for**
    **end for**
**end for**

---

# E   IMPLEMENT

In this section, we provide more experimental details. In our work, we utilize the CIFAR-10 (Krizhevsky & Hinton, 2009), CIFAR-100 (Krizhevsky & Hinton, 2009), and Tiny-ImageNet-200 (Deng et al., 2009). In the testing process, the primary reason for selecting FGSM (Madry et al., 2018), PGD (Carlini et al., 2019), BIM (Goodfellow et al., 2015), MIM (Dong et al., 2018), CW(Carlini & Wagner, 2017) as attack methods is to keep consistent with the baseline methods from the literature. Further, we select AA (Croce & Hein, 2020) because it is also a popular attack method and more powerful than those base methods. To reduce the computational complexity of targeted attacks, we leverage the transferability of adversarial examples and utilize a pre-trained simple network (VGG11(Simonyan & Zisserman, 2015)) structure for targeted attacks.

Further, we introduce the implement of "FDT-random", "FDT-target" and "FDT-hybrid" here. For "FDT-random", we perform Fourier Transform on $x$ and also on a randomly sampled standard Gaussian noise $n_0$. Then, we can obtain the low-amplitude frequencies and high-amplitude frequencies of $x$ by setting an amplitude threshold. Next, we generate two masks ($M_1$ and $M_2$) to select high-amplitude frequencies and low-amplitude frequencies. We add the low-amplitude frequencies of $n_0$ (i.e., $M_2(n_0)$) to high-amplitude frequencies of $x$ (i.e., $M_1(x)$), and obtain the transformation of $x$ (denoted as $\pi(x)$). Finally, we transform $\pi(x)$ to time domain by inverse Fourier transform and train the model with $\pi(x)$. For "FDT-target", we obtain the targeted class according to the allocation scheme mentioned in "Stage (1)". Then, we use targeted PGD attack to obtain the adversarial sample $x'$. After that, perform the same steps as with FDT-random (we substitute $n_0$ with $x'$). For FDT-hybrid, we set two frequency selection thresholds $\tau_1$ and $\tau_2$ ($\tau_1 < \tau_2$), and generate three masks to select the frequencies: $M_1$ for the high-amplitude frequencies (amplitude $> \tau_2$ ), $M_2$ for the middle part ($\tau_1 <$ amplitude $< \tau_2$ ), and $M_3$ for the small part (amplitude $< \tau_1$). Next, we combine $M_1(x), M_2(x')$ and $M_3(n_0)$ to obtain the transformation $\pi(x)$.

# F   ADDITIONAL EXPERIMENTAL RESULTS

In this section, we provide more experimental results. Firstly, we extend our experiments to SVHN, Tiny-ImageNet-200, and WideResNet-28-10 in Appendix F.1. We also conduct the ablation studies on weakness set allocation method, amplitude-based selection threshold and model architecture in Appendix F.1. Then, we evaluate the performance of FDT under black-box attacks on the CIFAR-10 and CIFAR-100 in Appendix F.2 Then we present the trade-off between clean accuracy and robust accuracy on the CIFAR-100 using FDT method in Appendix F.3. This trade-off sheds light on the effectiveness of FDT with changing the trade-off parameter. Additionally, in Appendix F.4, we compare the transferability across various sub-models with the baseline methods. Furthermore, we compare our method with more related methods in Appendix F.5.

## F.1 ABLATION STUDIES

In this section, we extend our experiments to additional datasets (SVHN, Tiny-ImageNet-200) and architecture (WideResNet-28-10). We also explore the ablation studies on weakness set allocation method, amplitude-based selection threshold and model architecture.

Table 3 presents the performance of ensemble methods trained with ResNet-20 on SVHN against several widely used white-box attacks. The experimental results demonstrate that all ensemble models achieve comparable levels of clean accuracy. Specifically, the FDT approach exhibits better robust accuracy than the other methods. These observations highlight the effectiveness of FDT in achieving favorable clean accuracy and robustness of ensemble models.

Table 3: Robust Accuracy (%) of different ensemble methods against white-box attacks on SVHN. The $\epsilon$ and $\lambda$ stand for the $l_\infty$ norm of the adversarial perturbation and the coefficient of C&W attack respectively. The last column is the ensemble model trained with FDT-hybrid.

| **SVHN** | ADP | GAL | DVERGE | TRS | **FDT-hybrid** |
|---|---|---|---|---|---|
| clean accuracy | **96.83** | 94.66 | 96.28 | 94.52 | 96.73 $\pm$ 0.12 |
| FGSM ($\epsilon$=0.01) | 84.38 | 80.2 | 85.6 | 72.87 | **90.13** $\pm$ 0.09 |
| FGSM ($\epsilon$=0.02) | 78.08 | 41.5 | 81.4 | 53.9 | **86.78** $\pm$ 0.07 |
| PGD ($\epsilon$= 0.01) | 51.01 | 50.1 | 53.31 | 54.43 | **59.42** $\pm$ 0.07 |
| PGD ($\epsilon$ = 0.02) | 17.74 | 8.24 | 17.42 | 18.86 | **22.74** $\pm$ 0.04 |
| BIM ($\epsilon$= 0.01) | 54.38 | 47.73 | 52.08 | 53.71 | **57.91** $\pm$ 0.08 |
| BIM ($\epsilon$ = 0.02) | **21.26** | 8.1 | 14.58 | 18.05 | 20.23 $\pm$ 0.05 |
| MIM ($\epsilon$= 0.01) | 61.24 | 51.96 | 58.51 | 56.32 | **62.14** $\pm$ 0.08 |
| MIM ($\epsilon$= 0.02) | 24.84 | 5.14 | 23.22 | 21.95 | **25.37** $\pm$ 0.04 |
| AA ($\epsilon$= 0.01) | 49.92 | 48.39 | 52.02 | 52.83 | **57.54** $\pm$ 0.09 |
| AA ($\epsilon$= 0.02) | 16.13 | 6.90 | 16.95 | 17.48 | **20.12** $\pm$ 0.05 |
| C&W ($\lambda$ = 0.1) | 55.81 | 49.94 | 66.82 | 52.74 | **72.14** $\pm$ 0.11 |

We also extend our experiment to the sub-models trained with WideResNet-28-10 on CIFAR-10. Table 4 shows the performance of the models facing various whitebox attacks. The results indicate that FDT maintains good performance even on more complex network structures. We also evaluated the robustness of an ensemble of eight sub-models, with the results presented in Table 5.

Table 4: Robust Accuracy (%) of different ensemble methods against white-box attacks on CIFAR-10. The $\epsilon$ and $\lambda$ stand for the $l_\infty$ norm of the adversarial perturbation and the coefficient of C&W attack respectively. The architecture of sub-model is WRN-28-10.

| **CIFAR-10** | ADP | GAL | DVERGE | **FDT-hybrid** |
|---|---|---|---|---|
| clean accuracy | 92.99 | 82.14 | **94.32** | 94.18 $\pm$ 0.06 |
| FGSM ($\epsilon$=0.01) | 60.04 | 44.94 | 71.01 | **80.64** $\pm$ 0.05 |
| FGSM ($\epsilon$=0.02) | 51.69 | 36.83 | 50.43 | **60.09** $\pm$ 0.05 |
| PGD ($\epsilon$= 0.01) | 11.09 | 22.10 | 44.25 | **64.64** $\pm$ 0.07 |
| PGD ($\epsilon$ = 0.02) | 2.54 | 5.06 | 13.27 | **26.0** $\pm$ 0.03 |
| BIM ($\epsilon$= 0.01) | 15.81 | 22.62 | 46.53 | **67.36** $\pm$ 0.10 |
| BIM ($\epsilon$ = 0.02) | 4.50 | 5.43 | 17.38 | **32.36** $\pm$ 0.06 |
| MIM ($\epsilon$= 0.01) | 18.18 | 25.97 | 44.21 | **64.36** $\pm$ 0.08 |
| MIM ($\epsilon$= 0.02) | 4.72 | 7.81 | 12.83 | **25.64** $\pm$ 0.05 |
| AA ($\epsilon$= 0.01) | 9.38 | 19.34 | 43.23 | **63.45** $\pm$ 0.08 |
| AA ($\epsilon$= 0.02) | 1.17 | 3.93 | 12.49 | **25.23** $\pm$ 0.04 |
| C&W ($\lambda$ = 0.1) | 37.81 | 19.05 | 46.32 | **47.23** $\pm$ 0.10 |

Table 6 is the result of ensemble methods trained with WideResNet-28-10 on Tiny-ImageNet-200. We test the robustness of different methods under widely used white-box attacks. Due to the high

Table 5: Robust Accuracy (%) of an ensemble of eight sub-models against white-box attacks on CIFAR-10. The $\epsilon$ and $\lambda$ stand for the $l_\infty$ norm of the adversarial perturbation and the coefficient of C&W attack respectively. The architecture of sub-model is WRN-28-10.

| CIFAR-10 | FDT-hybrid |
|---|---|
| clean accuracy | $93.72 \pm 0.11$ |
| FGSM ($\epsilon$=0.01) | $86.31 \pm 0.07$ |
| FGSM ($\epsilon$=0.02) | $67.29 \pm 0.06$ |
| PGD ($\epsilon$= 0.01) | $72.02 \pm 0.07$ |
| PGD ($\epsilon$ = 0.02) | $45.42 \pm 0.05$ |
| BIM ($\epsilon$= 0.01) | $73.68 \pm 0.10$ |
| BIM ($\epsilon$ = 0.02) | $44.53 \pm 0.06$ |
| MIM ($\epsilon$= 0.01) | $71.36 \pm 0.06$ |
| MIM ($\epsilon$= 0.02) | $45.24 \pm 0.06$ |
| AA ($\epsilon$= 0.01) | $70.45 \pm 0.08$ |
| AA ($\epsilon$= 0.02) | $44.23 \pm 0.07$ |
| C&W ($\lambda$ = 0.1 | $72.37 \pm 0.11$ |

time complexity of the TRS, we do not compare with it here. The experimental results show that all ensemble models achieve comparable levels of clean accuracy while FDT-hybrid achieves better robust accuracy than other methods.

Table 6: Robust Accuracy (%) of different ensemble methods against white-box attacks on Tiny-ImageNet-200. The $\epsilon$ and $\lambda$ stand for the $l_\infty$ norm of the adversarial perturbation and the coefficient of C&W attack respectively. The last column is the ensemble model trained with FDT-hybrid.

| Tiny-ImageNet-200 | ADP | GAL | DVERGE | FDT-hybrid |
|---|---|---|---|---|
| clean accuracy | 49.88 | 45.7 | 51.46 | **64.21** $\pm$ 0.06 |
| FGSM ($\epsilon = 0.01$) | 10.46 | 1.24 | **22.82** | $21.73 \pm 0.04$ |
| FGSM ($\epsilon = 0.02$) | 4.38 | 0.59 | 18.42 | **19.28** $\pm$ 0.04 |
| PGD ($\epsilon = 0.01$) | 0.02 | 0.02 | 3.6 | **4.76** $\pm$ 0.02 |
| PGD ($\epsilon = 0.02$) | 0.02 | 0.01 | 0.34 | **0.45** $\pm$ 0.01 |
| BIM ($\epsilon = 0.01$) | 0.07 | 0.02 | 3.35 | **4.81** $\pm$ 0.03 |
| BIM ($\epsilon = 0.02$) | 0.03 | 0.01 | 0.28 | **0.32** $\pm$ 0.00 |
| MIM ($\epsilon = 0.01$) | 0.11 | 0.02 | 4.36 | **6.13** $\pm$ 0.03 |
| MIN ($\epsilon = 0.02$) | 0.03 | 0.01 | 0.41 | **0.48** $\pm$ 0.00 |
| AA ($\epsilon = 0.01$) | 0 | 0 | 0 | **2.66** $\pm$ 0.02 |
| AA ($\epsilon = 0.02$) | 0 | 0 | 0 | **0.02** $\pm$ 0.00 |
| CW ($\lambda = 0.01$) | 2.36 | 0.13 | 9.54 | **19.47** $\pm$ 0.06 |

**Ablation study on model architectures.** Table 7 presents the results across different model architectures, including ResNet20, ResNet50, WRN28-10, and WRN34-10. While larger models generally achieve higher clean and robust accuracy, the results suggest that our method consistently enhances robustness under various attack scenarios, demonstrating its applicability across diverse architectures.

**Ablation study on allocation methods.** Table 8 compares the performance of FDT-hybrid with different weakness set allocation methods on CIFAR-10. The results indicate that our proposed allocation method achieves better clean accuracy and robustness under various attack scenarios than randomly uniform allocation.

**Ablation study on $\tau_1$ and $\tau_2$.** Table 9 presents the results of FDT-hybrid with various combinations of selection thresholds $\tau_1$ and $\tau_2$ on CIFAR-10. The experiments reveal the impact of different thresholds on both clean accuracy and robustness under adversarial attacks. As $\tau_2$ increases, robustness improves across all metrics, but clean accuracy decreases. For a fixed $\tau_2$, increasing $\tau_1$ generally leads to a trade-off between clean accuracy and robustness. Setting $\tau_1 = 0.2$ and $\tau_2 = 0.8$ achieves a relatively

balanced performance, maintaining both competitive clean accuracy and robust accuracy under various attacks.

Table 7: Robust Accuracy (%) of different model architectures against white-box attacks on Cifar10. The $\epsilon$ and $\lambda$ stand for the $l_\infty$ norm of the adversarial perturbation and the coefficient of C&W attack respectively.

| CIFAR10 | ResNet20 | ResNet50 | WRN28-10 | WRN34-10 |
|---|---|---|---|---|
| clean accuracy | 90.02 | 93.23 | 94.18 | 94.63 |
| FGSM ($\epsilon = 0.01$) | 72.24 | 76.65 | 80.64 | 81.04 |
| FGSM ($\epsilon = 0.02$) | 58.04 | 58.59 | 60.09 | 60.92 |
| PGD ($\epsilon = 0.01$) | 48.48 | 60.23 | 64.64 | 65.38 |
| PGD ($\epsilon = 0.02$) | 20.01 | 24.35 | 26.00 | 27.42 |
| BIM ($\epsilon = 0.01$) | 48.57 | 60.43 | 67.36 | 68.29 |
| BIM ($\epsilon = 0.02$) | 16.63 | 23.57 | 32.36 | 33.86 |
| MIM ($\epsilon = 0.01$) | 51.48 | 60.81 | 64.36 | 64.71 |
| MIN ($\epsilon = 0.02$) | 20.09 | 24.54 | 25.64 | 26.42 |
| AA ($\epsilon = 0.01$) | 51.56 | 60.48 | 63.45 | 64.01 |
| AA ($\epsilon = 0.02$) | 19.42 | 24.21 | 25.23 | 26.39 |
| CW ($\lambda = 0.01$) | 56.08 | 56.55 | 57.23 | 57.52 |

Table 8: Performance of FDT-hybrid with different weakness set allocation method on CIFAR-10. The other settings are consistent with those in Table 1.

| Allocation method | Clean accuracy | FGSM ($\epsilon = 0.02$) | PGD ($\epsilon=0.02$) | AutoAttack ($\epsilon=0.02$) |
|---|---|---|---|---|
| Uniform Random | 89.32 | 56.20 | 18.24 | 17.89 |
| Ours | 90.20 | 58.04 | 20.01 | 19.42 |

Table 9: Performance of FDT-hybrid with different selection thresholds $\tau_1$ and $\tau_2$ on CIFAR-10. The other settings are consistent with those in Table 1.

| Thresholds | Clean accuracy | FGSM ($\epsilon = 0.02$) | PGD ($\epsilon=0.02$) | AutoAttack ($\epsilon=0.02$) |
|---|---|---|---|---|
| $\tau_1 = 0.2, \tau_2 = 0.7$ | 91.03 | 56.62 | 17.74 | 17.60 |
| $\tau_1 = 0.2, \tau_2 = 0.8$ | 90.20 | 58.04 | 20.01 | 19.42 |
| $\tau_1 = 0.2, \tau_2 = 0.9$ | 89.46 | 58.48 | 20.12 | 19.57 |
| $\tau_1 = 0.4, \tau_2 = 0.7$ | 89.75 | 56.89 | 17.93 | 17.82 |
| $\tau_1 = 0.4, \tau_2 = 0.8$ | 89.08 | 58.21 | 20.01 | 19.47 |
| $\tau_1 = 0.4, \tau_2 = 0.9$ | 88.44 | 58.53 | 20.09 | 19.61 |
| $\tau_1 = 0.6, \tau_2 = 0.7$ | 89.62 | 53.35 | 15.27 | 14.63 |
| $\tau_1 = 0.6, \tau_2 = 0.8$ | 88.84 | 55.33 | 15.42 | 15.24 |
| $\tau_1 = 0.6, \tau_2 = 0.9$ | 88.12 | 55.46 | 15.83 | 15.47 |

## F.2 RESULTS FOR BLACK-BOX ATTACK

In the black-box setting, the attacker's knowledge usually is limited to the original training dataset and has no information about the model. This setting represents a more practical attack scenario. The attacker can train a surrogate model to generate transferable adversarial examples and transfer them to the target ensemble model. We utilize a single ResNet-20 model as the surrogate model. Adversarial examples are generated on the surrogate model using the SPSA algorithm (Spall, 1992). Figure 5 shows the robust accuracy of ensemble models against black-box attacks under different degrees of perturbation. As we can see, FDT-hybrid ensemble training strategies outperform the other ensemble training strategy against black-box attacks both on CIFAR10 and CIFAR100.

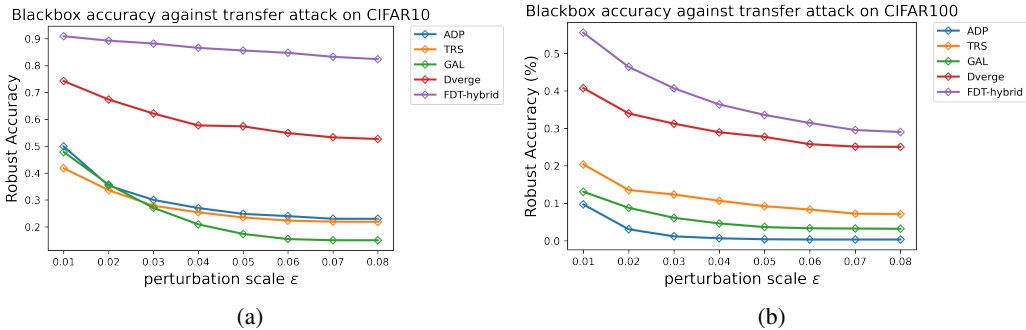

(a)               (b)

Figure 5: Robust Accuracy for different ensemble models against black-box attack with different perturbation scale $\epsilon$.

## F.3 TRADE-OFF BETWEEN CLEAN AND ROBUST ACCURACY

In this section, we explore the trade-off between clean accuracy and robust accuracy by varying the frequency selection threshold $\tau_2$ (as mentioned in Section 4.2). And we set $\tau_1$ to be 0.1. To assess the adversarial robustness, we utilize the PGD attack under $l_\infty$ perturbations of size $\epsilon = 0.01$ as a benchmark. We train a set of ResNet-20 FDT-hybrid models on CIFAR-10 and CIFAR-100 with various frequency selection threshold $\tau_2 \in \{0.4, 0.6, 0.8, 1.0, 1.2, 1.6\}$. Figure 6 shows that the ensemble model has lower clean accuracy and higher robust accuracy with the increasing of $\tau_2$.

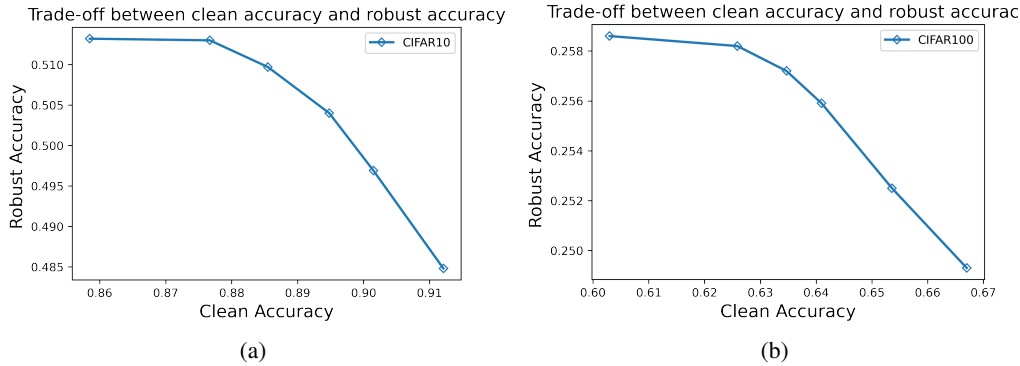

(a)               (b)

Figure 6: (a) shows the trade-off on CIFAR-10 while (b) on CIFAR-100. From left to right, we decrease the trade-off parameter $\tau_2$ for FDT.

## F.4 TRANSFERABILITY ACROSS VARIOUS SUB-MODELS

To further investigate the diversity between sub-models, we conduct an analysis by generating adversarial examples using one sub-model and evaluating their accuracy on other target sub-models. The transferability of these adversarial examples among sub-models is visualized in Figure 7, considering different ensemble training methods on the CIFAR10 dataset. We generate adversarial examples from "base model" and test the accuracy of "target model" .The experimental results indicate that FDT exhibits comparable performance to DVERGE and TRS in reducing the transferability of adversarial samples across different sub-models. This demonstrates that FDT not only enhances the diversity of weaknesses within the dataset but also weakens the transferability of adversarial examples between sub-models.

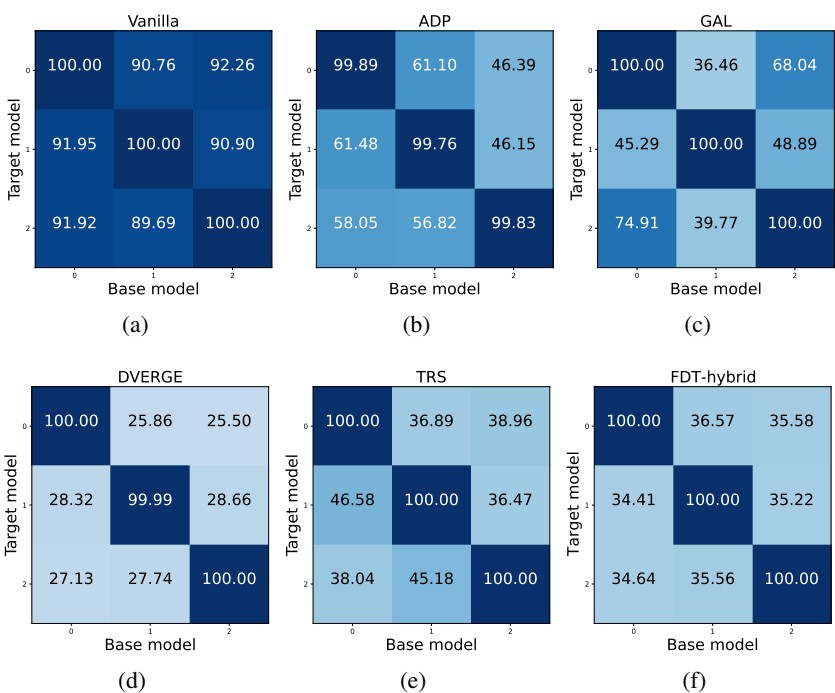

Figure 7: Pair-wise adversarial transferability between sub-models against PGD attack with $\epsilon = 0.02$ on CIFAR-10. The value represents the success rate of adversarial examples generated by the base model in attacking the target model.

### F.5 COMPARE WITH ADVERSARIAL TRAINING

We use target attacks in our data transformation, which differs significantly from adversarial training. First, we employ a simple pre-trained network (VGG11 in our experiment) to compute adversarial examples, thereby accelerating the training process. Second, we only utilize the low amplitude part of the adversarial examples for data transformation, which helps maintain the model's clean accuracy. We compare our method with several popular approaches (Wang et al., 2023b; Rade & Moosavi-Dezfooli, 2021; Xu et al., 2023) on CIFAR-10 using AutoAttack under $l_\infty$ perturbations ($\epsilon = 8/255$). Wang et al. (2023b) generated training datasets using a diffusion model, followed by adversarial training on these datasets. For fairness, we compare our method with the version proposed by Wang et al. (2023b) that uses 50k generated images. Rade & Moosavi-Dezfooli (2021) used "helper example" to help the adversarial training. Xu et al. (2023) proposed Dynamics-Aware Robust Training, which encourages the decision boundary to adjust in a way that prioritizes increasing smaller margins. We use WideResNet-28-10 as the sub-model and ensemble eight sub-models without using generated data. The results in Table 10 indicate that, although the robustness of our method is not the highest, it maintains clean accuracy with almost no decline. Moreover, our method does not require additional generated data or adversarial training, and even with the need for ensembling, the training efficiency remains relatively high. This suggests a potential way to enhance robustness while minimizing the decrease in clean accuracy.

To further illustrate our method's advantage, we conduct additional experiments to compare the "robustness-clean accuracy" trade-off curves of our method and AT under different settings. Fig. 8 compares the trade-off curves obtained by HAT Rade & Moosavi-Dezfooli (2021) with that of FDT-hybrid. For HAT, we fix $\gamma = 0.25$ and vary $\beta \in \{0.1, 0.5, 2.5, 3.0, 4.0, 5.0\}$ ($\beta$ is the coefficient of the robustness loss, and higher $\beta$ indicates higher robust accuracy); for FDT-hybrid, we fix $\tau_1 = 0.2$ and vary $\tau_2 \in \{0.5, 0.7, 0.9, 1.1, 1.3, 1.5\}$. We observe that HAT's robustness declines rapidly when the $\beta$ parameter is small (as increasing the clean accuracy). This result shows the significant advantage of our method when a clean accuracy above 90% is required.

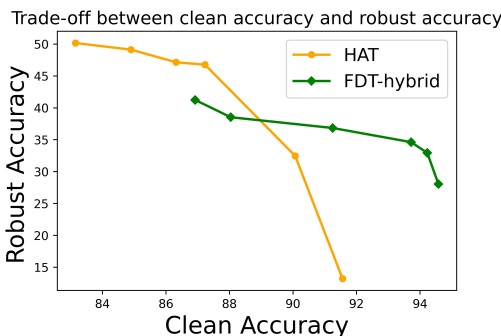

Figure 8: It shows the trade-off curves on CIFAR-10. From left to right, we decrease the trade-off parameter $\tau_2$ for FDT, and decrease the trade-off parameter $\beta$ for HAT.

Table 10: Clean accuracy and robust accuracy (%) of different methods against AutoAttack under $l_\infty$ perturbations ($\epsilon = 8/255$) on CIFAR-10.

| CIFAR-10 | clean accuracy | robust accuracy |
|---|---|---|
| (Wang et al., 2023b) | 86.15 | **55.71** |
| (Rade & Moosavi-Dezfooli, 2021) | 84.90 | 49.08 |
| (Xu et al., 2023) | 85.55 | 54.69 |
| **OURS (FDT-hybrid)** | **93.72** | 34.61 |

