# OpenReview forum: "To Tackle Adversarial Transferability: A Novel Ensemble Training Method with Fourier Transformation"
_ICLR.cc/2025/Conference — ICLR 2025 Poster_

### Official Review · Reviewer_QFba · 2024-11-02

**Soundness:** 2
**Presentation:** 3
**Contribution:** 2
**Rating:** 6
**Confidence:** 4

**Summary:**

This paper introduces a data transformation method aimed at improving the adversarial robustness of ensemble models. Instead of reducing the influence of non-robust features, the authors propose to reduce the transferability of attacks by increasing the diversity of non-robust features. The robust and non-robust features are identified by frequency selection. The weakness set is allocated to sub-models for training to increase diversity. Through the experiments on various datasets, the results demonstrate the superiority of the proposed method over other baselines.

**Strengths:**

1. This paper is well-written and easy to follow.
2. The motivation of diverse non-robust features seems natural for a better trade-off between clean and robust accuracy.
3. The comparison with other baselines shows the effectiveness of the proposed method.

**Weaknesses:**

1. The motivation of amplitude-based selection is not well explained. There is no theoretical or empirical evidence to support this design in this paper.
2. There is no ablation study to verify the effectiveness of all the components, such as weakness set allocation, amplitude-based selection, new dataset construction, etc.
3. The results of baselines seem strange. For example, the results in Table 2 differ from those in Table 1 of TRS.

**Questions:**

1. Please clarify the motivation of amplitude-based selection.
2. Please provide more ablation studies of the proposed method.
3. Please clarify the difference in the result in Table 2.

---

> ### Author Response · Authors · 2024-11-23
> **Rebuttal**
>
> **Please clarify the motivation of amplitude-based selection.**
>
> We thank the reviewer for the question. Due to space constraints, we have moved the motivation  of amplitude-based selection to Appendix B (line 791-795). In Appendix B, we use an example to show that, if we keep  high-amplitude frequencies and remove low-amplitude ones, the image is changed slightly even with adding certain noise (i.e., we can still recognize the ground truth from the modified image). On the other hand, if we keep the low-amplitude frequencies only, the semantic information is almost missing. This observation suggests that high-amplitude frequency features are more strongly related to the semantic information of image. Through amplitude selection, we retain high-amplitude components and transform low-amplitude ones, thereby preserving clean accuracy as much as possible.
>
> **Please provide more ablation studies of the proposed method.**
>
> Thank you for raising this valuable question. We have conducted ablation studies on the number of sub-models (Table 1), datasets (Tables 2 and 3), and model architectures (Tables 2 and 4) in the original manuscript, which we hope address some aspects of your concerns.
>
> Additionally, we have now included further ablation studies on weakness set allocation and amplitude-based selection in response to your feedback. The first table compares the performance of FDT-hybrid with different weakness set allocation methods on CIFAR-10. The results indicate that our proposed allocation method achieves better clean accuracy and robustness under various attack scenarios than randomly uniform allocation. The second table presents the results of FDT-hybrid with various combinations of selection thresholds $\tau_1$ and $\tau_2$ on CIFAR-10.
>
> | **Allocation Method** | Clean Accuracy | FGSM ($\epsilon=0.02$) | PGD ($\epsilon=0.02$) | AutoAttack ($\epsilon=0.02$) |
> | --------------------- | -------------- | ---------------------- | --------------------- | ---------------------------- |
> | Uniform Random        | 89.32          | 56.20                  | 18.24                 | 17.89                        |
> | Ours                  | 90.20          | 58.04                  | 20.01                 | 19.42                        |
>
>
>
> | Threshold               | Clean Accuracy | FGSM ($\epsilon=0.02$) | PGD ($\epsilon=0.02$) | AutoAttack ($\epsilon=0.02$) |
> | ----------------------- | -------------- | ---------------------- | --------------------- | ---------------------------- |
> | $\tau_1=0.2,\tau_2=0.7$ | 91.03          | 56.62                  | 17.74                 | 17.60                        |
> | $\tau_1=0.2,\tau_2=0.8$ | 90.20          | 58.04                  | 20.01                 | 19.42                        |
> | $\tau_1=0.2,\tau_2=0.9$ | 89.46          | 58.48                  | 20.12                 | 19.57                        |
> | $\tau_1=0.4,\tau_2=0.7$ | 89.75          | 56.89                  | 17.93                 | 17.82                        |
> | $\tau_1=0.4,\tau_2=0.8$ | 89.08          | 58.21                  | 20.01                 | 19.47                        |
> | $\tau_1=0.4,\tau_2=0.9$ | 88.44          | 58.53                  | 20.09                 | 19.61                        |
> | $\tau_1=0.6,\tau_2=0.7$ | 89.62          | 53.35                  | 15.27                 | 14.63                        |
> | $\tau_1=0.6,\tau_2=0.8$ | 88.84          | 55.33                  | 15.42                 | 15.24                        |
> | $\tau_1=0.6,\tau_2=0.9$ | 88.12          | 55.46                  | 15.83                 | 15.47                        |
>
>
> **Please clarify the difference in the result in Table 2.**
>
> Thank you for the question. We would like to clarify that Table 1 reports the results of FDT-hybrid with varying numbers of sub-models. It does not appear to include the results for TRS.

---

> > ### Comment · Reviewer_QFba · 2024-11-25
> >
> > I thank the authors for their response. However, I still have concerns regarding the results of Table 2. The reported TRS results in Table 2 are different from those in Table 1 of TRS paper [a]. Would you please clarify the difference here?
> >
> > [a]. TRS: Transferability Reduced Ensemble via Promoting Gradient Diversity and Model Smoothness.

---

> > > ### Author Response · Authors · 2024-11-27
> > >
> > > Thank you for your  valuable feedback. We follow the parameters provided in the publicly available code of TRS  [https://github.com/AI-secure/Transferability-Reduced-Smooth-Ensemble](https://github.com/AI-secure/Transferability-Reduced-Smooth-Ensemble). The key differences are that we decay the learning rate by 0.1 every 40 epochs, whereas TRS paper applies decay at the 100th and 150th epochs, and we use the coefficients $ L_{std} = 1,L_{sim} = 10,L_{smooth} = 10$ from that TRS code. Morevover, the used parameters corresponding to Table 1 in the original TRS paper is not released in their paper. We also would like to mention that, the TRS results reproduced in our implementation are  close to those reproduced by some recent published papers, such as MORA [1].
> > >
> > > Nevertheless, to better address your concern, we have directly adopted the results reported in the TRS paper and revised Table 1 in our manuscript accordingly (this change does not affect to show our method's advantage in both clean accuracy and robust accuracy).
> > >
> > > [1]Xitong Gao, Cheng-Zhong Xu, et al. Mora: Improving ensemble robustness evaluation with model reweighing attack. Advances in Neural Information Processing Systems, 35:26955–26965, 2022.

---

### Official Review · Reviewer_XA1w · 2024-11-04

**Soundness:** 3
**Presentation:** 3
**Contribution:** 3
**Rating:** 6
**Confidence:** 2

**Summary:**

This paper proposes an ensemble training approach to tackle the adversarial transferability problem by leveraging Fourier transformation and a weakness allocation strategy to diversify non-robust features across sub-models. The authors introduce a novel data transformation framework involving frequency selection and frequency transformation, aiming to improve the ensemble model’s robustness against adversarial attacks without sacrificing clean accuracy. Experimental results show that their method, particularly the FDT-hybrid, outperforms several existing ensemble methods in robust accuracy on benchmark datasets like CIFAR-10 and CIFAR-100​.

**Strengths:**

1. The paper addresses a critical challenge in adversarial machine learning, focusing on enhancing ensemble robustness without excessive overhead.
2. The frequency-based approach for diversifying non-robust features is innovative and leverages insights from signal processing.
3. Experimental results are thorough, showcasing improved robustness across various adversarial attacks and comparison with multiple baseline methods.

**Weaknesses:**

1. The paper’s complexity may hinder reproducibility, especially given the intricate weakness allocation and frequency transformation processes.
2. The dependency on specific frequency thresholding could limit adaptability across different datasets or tasks.

**Questions:**

1. Could the authors elaborate on how the choice of frequency thresholds (τ1 and τ2) affects robustness and clean accuracy?
2. Is the computational efficiency of the proposed approach scalable to larger models or more complex datasets, given the Fourier transformations applied?

---

> ### Author Response · Authors · 2024-11-23
> **Rebuttal**
>
> **Question1**
>
> Thanks for the question. During our parameter tuning process, we observed that increasing $\tau_2$ leads to lower clean accuracy but higher robustness. When $\tau_2$  is fixed, increasing  $\tau_1$ similarly decreases clean accuracy while improving robustness. However, if  $\tau_1$ becomes excessively large, both clean accuracy and robustness deteriorate. The following table presents the results of FDT-hybrid with various combinations of selection thresholds $\tau_1$ and $\tau_2$ on CIFAR-10. So based on these exeperimental observations,  we setting  $\tau_1=0.2$ and  $\tau_2=0.8$ to achieve a relatively balanced performance, maintaining both competitive clean accuracy and robust accuracy under various attacks.
>
> | Threshold               | Clean Accuracy | FGSM ($\epsilon=0.02$) | PGD ($\epsilon=0.02$) | AutoAttack ($\epsilon=0.02$) |
> | - | - | - | - | - |
> | $\tau_1=0.2,\tau_2=0.7$ | 91.03          | 56.62                  | 17.74                 | 17.60                        |
> | $\tau_1=0.2,\tau_2=0.8$ | 90.20          | 58.04                  | 20.01                 | 19.42                        |
> | $\tau_1=0.2,\tau_2=0.9$ | 89.46          | 58.48                  | 20.12                 | 19.57                        |
> | $\tau_1=0.4,\tau_2=0.7$ | 89.75          | 56.89                  | 17.93                 | 17.82                        |
> | $\tau_1=0.4,\tau_2=0.8$ | 89.08          | 58.21                  | 20.01                 | 19.47                        |
> | $\tau_1=0.4,\tau_2=0.9$ | 88.44          | 58.53                  | 20.09                 | 19.61                        |
> | $\tau_1=0.6,\tau_2=0.7$ | 89.62          | 53.35                  | 15.27                 | 14.63                        |
> | $\tau_1=0.6,\tau_2=0.8$ | 88.84          | 55.33                  | 15.42                 | 15.24                        |
> | $\tau_1=0.6,\tau_2=0.9$ | 88.12          | 55.46                  | 15.83                 | 15.47                        |
>
> **Question2**
>
> We thank the reviewer for the valuable question.  We believe that our approach is scalable to larger models or more complex datasets. The primary computational overhead of our method comes from the Fourier transform and the extraction of adversarial features. The computational complexity of FFT is  $nlog(n)$ (n is the image dimension), which is acceptable compared to the complexity of neural network convolutions. For the adversarial features, our proposed approach is a general framework that can be incorporated with any attack transformation (so it is quite flexible in practice). Based on the transferability of adversarial examples, we can utilize a simple network architecture as pre-trained model to perform targeted attack transformation to obtain these features, and consequently the total computational cost is not increased too much. As described in Appendix E (line 894), we utilized VGG11 in our experiments. Another advantage that enables our model to scale easily to larger datasets and models is that our sub-models can be trained independently without requiring communication.
>
> **Weakness1**
>
> We appreciate your comment. To address the concern,we would like to clarify that we applied our method to various datasets, including CIFAR-10, CIFAR-100, SVHN, and Tiny-ImageNet, as well as different models such as ResNet-20 and WRN28-10. Reproducing our approach on new tasks primarily involves adjusting the corresponding parameters, suggesting that the algorithm can be reproduced with reasonable effort. We have submitted our code and will make it publicly available to support other researchers in reproducing our work.
>
> **Weakness2**
>
> We thank the reviewer for the thoughtful comment. We agree with the reviewer that the thresholds are dataset-dependent. However, we believe this does not significantly limit the adaptability of our method across different datasets or tasks. We would like to provide a method for determining initial values of $\tau_1$ and $\tau_2$ on new datasets based on our experience with CIFAR-10, CIFAR-100, SVHN, and Tiny-ImageNet. When faced with a new dataset, we can randomly select a subset of samples, perform Fourier transforms on them, and sort the amplitude values. We then choose the lower 30\% threshold and the upper 30\% threshold as the initial values for $\tau_1$ and $\tau_2$, respectively. Based on our existing experiments, we consider 30\% to be a reasonable balance point. Then, we can fine-tune the model based on the influence of $\tau_1$ and $\tau_2$ as mentioned in Question1.

---

> ### Author Response · Authors · 2024-11-28
>
> Dear Reviewer XA1w,
>
> We would like to sincerely thank you once again for your thoughtful comments and valuable suggestions on our work. As mentioned in our previous response, we have revised the manuscript based on your feedback and hope that these changes and our responses effectively address your concerns. If you have any further comments, we would be more than happy to address them.
>
> Sincerely,
>
> The authors of Paper#9133

---

### Official Review · Reviewer_hPDX · 2024-11-09

**Soundness:** 3
**Presentation:** 1
**Contribution:** 2
**Rating:** 6
**Confidence:** 3

**Summary:**

Ensemble training is a commonly used technique to enhance a model's adversarial robustness. However, recent studies suggest that this approach may not be as effective, as adversarial examples can often affect multiple sub-models due to a phenomenon called "transferability." This limitation arises because all sub-models are typically trained on the same dataset, leading them to share similar vulnerabilities. To address such issue, the author propose an effective data transformation framework to improve the diversity of training
data used by different sub-models for robust ensemble training. In the end of the paper, they present empirical results demonstrating the effectiveness of their method against common adversarial attacks, while also maintaining clean accuracy.

**Strengths:**

- The proposed framework is innovative, intuitively reasonable, and easy to understand.
- The method comes with solid theorical backing
- The experiments are comprehensive and well-supported their findings

**Weaknesses:**

- The paper's readability is poor. I’ve provided specific feedback in the following sections and strongly recommend improvements in clarity and structure.
- The ensemble learning with proposed method still shows limited competitiveness with adversarial training, which restricts its practical applicability. Although the appendix shows that this method maintains higher clean accuracy than popular adversarial training (AT) methods, it still falls significantly short in robust accuracy compared to these approaches.

**Questions:**

Here are few questions and suggestion I want to provided for your paper:

1. No detailed explanation of the feature extractor. In Definition 3.1, you provide a detailed definition of what a "useful detail extractor" is. However, the feature extractor itself is only given as $θ: X → \mathcal{R}^k$. Based on the information provided, I can't see how it differs from a standard classification model. I hope you can add more details to improve the readability of the paper

2. $\hat{y}$ is not an appropriate mathematical symbol in my opinion. $\hat{y}$ is typically used to denote the prediction of y by convention, but in your paper, you are using it to denote the one-hot vector of y, which is quite confusing.

3. In your Definition 3.2, I am confused by the second formula you provided. The first formula indicates that the feature extractor is robust if the expectation $ y^t \cdot \theta(A(x)) > \frac{1}{k} $, but the second formula does not contain $y^t$ at all. I'm not sure whether I misunderstood your formula or if there is a mistake here.

4. The format of your formula 8 doesn’t seem correct to me. The equation is broken in the middle, leaving part of it followed after the plain text and the other part in equation mode.

5. I am curious how you find the "high-amplitude frequency features" from your image. I think I am having this question since I didn't understand how your feature extractors work.

---

> ### Author Response · Authors · 2024-11-23
> **Rebuttal**
>
> **Questions:**
>
> **No detailed explanation of the feature extractor. In Definition 3.1, you provide a detailed definition of what a "useful detail extractor" is. However, the feature extractor itself is only given as $\theta: x \rightarrow \mathcal{R}^k$ . Based on the information provided, I can't see how it differs from a standard classification model. I hope you can add more details to improve the readability of the paper.**
>
> We thank the reviewer for the comment and suggestion. We would like to add more details. The definition of feature extractor follows the work [1]. The setting is that the soft-classification model $f(x,\beta)$ can be written as a combination of a set of feature extractors. Different feature extractors focus on different features.  Given input x, the model $f(x,\beta)$ and the feature extractor return different outputs.  The combination of outputs from feature extractors forms the model's output.
>
> [1] Ilyas et al.,  Adversarial examples are not bugs, they are features, NIPS 2019.
>
>
>
> **$\hat{y}$ is not an appropriate mathematical symbol in my opinion. $\hat{y}$ is typically used to denote the prediction of y by convention, but in your paper, you are using it to denote the one-hot vector of y, which is quite confusing.**
>
> Thank you for the suggestion. We have revised our manuscript accordingly.
>
> **In your Definition 3.2, I am confused by the second formula you provided. The first formula indicates that the feature extractor is robust if the expectation $ y^t \cdot \theta(A(x)) > \frac{1}{k} $, but the second formula does not contain $y^t$ at all. I'm not sure whether I misunderstood your formula or if there is a mistake here.**
>
> Thanks for the question. In fact, for images of the $i$-th class, the term $ [\hat{y}^T\theta(\mathcal{A}(x))]$  in Equation (1) of Definition 3.2 and  $[\theta(\mathcal{A}(x))]_i$ in Equation (2) have the same meaning. Both refer to the  $i$-th dimension of the output of $\theta(\mathcal{A}(x))$. We initially used the former to elaborate on the details of the definition, while the latter was adopted later for simplicity. To address your concern, we will adopt the simplified notation throughout.
>
> **The format of your sformula 8 doesn’t seem correct to me. The equation is broken in the middle, leaving part of it followed after the plain text and the other part in equation mode.**
>
> Thank you for your concern. Due to space constraints, we arranged Equation (8) this way. We have adjusted its formatting to improve clarity.
>
> **I am curious how you find the "high-amplitude frequency features" from your image. I think I am having this question since I didn't understand how your feature extractors work.**
>
> Thank you for raising this question. In the experiments, we perform a Fourier transform on the images and set a threshold in the frequency domain. Frequencies exceeding this threshold are identified as high-amplitude frequencies. By applying the inverse Fourier transform to these frequencies, we obtain the high-amplitude frequency features in time domain.
>
> **weaknesses:**
>
> **The ensemble learning with proposed method still shows limited competitiveness with adversarial training, which restricts its practical applicability. Although the appendix shows that this method maintains higher clean accuracy than popular adversarial training (AT) methods, it still falls significantly short in robust accuracy compared to these approaches.**
>
> Thank you for your valuable concern. While we acknowledge that our method may not achieve the highest robust accuracy compared to adversarial training (AT) methods, we believe that a method offering higher clean accuracy alongside reasonable robustness can still be meaningful and potentially practical for certain real-world applications, such as autonomous driving and medical diagnosis, where datasets typically consist of a majority of clean samples and only a minority of adversarial samples. Recently, there have been discussions on the importance of clean accuracy in real-world applications [2], aiming to find a better trade-off.   Assuming the task involves 10\% adversarial samples,  under the setting described in Table 7, the overall accuracy of our method can be calculated as  93.72\% * 90%+34.61% * 10%=87.809%. In comparison, the overall accuracy of adversarial training would be 86.15% * 90% + 55.71% * 10%=83.106%.  This demonstrates that while adversarial training achieves higher robust accuracy, our method still maintains competitive performance under this specific distribution.
>
> [2]Bai Y, Anderson B G, Kim A, et al. Improving the accuracy-robustness trade-off of classifiers via adaptive smoothing[J]. SIAM Journal on Mathematics of Data Science, 2024, 6(3): 788-814

---

> ### Comment · Reviewer_hPDX · 2024-11-25
>
> I am appreciates the authors response and their efforts on reworking their manuscript. I still have following concerns:
>
> 1. Thanks for clarifying what is feature extractor here. However, it seems like you didn't add the details in your manuscript. I briefly reviewed the referenced paper [1] and gained more insight into the concept of a feature extractor from their work. In my view, your paper should include sufficient background information—such as a clear definition of a feature extractor and its role in classification tasks—similar to what [1] provides, as their work forms the foundational basis of your study. I personally believe it is unreasonable to expect readers to jump into all cited references to understand your work.
>
> 2. I personally agreed that the clean accuracy alongside reasonable robustness might be applicable for some specific applications. However, this statement does not seem to effectively support the applicability of your work. As noted in your appendix, your method exhibits approximately a 20% drop in robustness accuracy while achieving only a 7% improvement in clean accuracy compared to the competitor with the highest robustness accuracy using AT. Furthermore, AT could similarly prioritize clean accuracy at the expense accuracy of robustness if such a trade-off were explicitly allowed. Is there any evidence to demonstrate that your method achieves a superior robustness-clean accuracy trade-off compared to AT?
>
> Due to these concerns, I will still stay with the current score.

---

> ### Author Response · Authors · 2024-11-27
>
> Thank you for your thoughtful comments and detailed suggestions. We greatly appreciate your time and effort in reviewing our manuscript. Below, we  address your concerns point by point.
>
> 1. In response to your suggestion, we have updated our revised paper and included a more clear description for  feature extractor and its role in classification tasks in lines 154-157.
>
> 2. To further illustrate the advantage of our method, we conducted additional experiments comparing the ``robustness-clean accuracy'' trade-off curves of our method and AT under various settings. We use HAT [3], which is a recent and representative adversarial training (AT) approach. The results of these experiments have been included in Appendix F6 (marked in red). We observe that HAT's robustness declines rapidly when increasing the clean accuracy. Additionally, the results show that our method achieves a more favorable trade-off when the clean accuracy exceeds 90\%. We would like to note that, upon receiving your feedback, we promptly began experiments on 8 A100 GPUs. Due to the significant time required, we focused on comparisons with HAT for now.  In the final version of the manuscript, we will include comparisons with additional AT methods.
>
> [3]Rahul Rade and Seyed-Mohsen Moosavi-Dezfooli. Helper-based adversarial training: Reducing
> excessive margin to achieve a better accuracy vs. robustness trade-off. ICML 2021

---

> > ### Comment · Reviewer_hPDX · 2024-11-28
> >
> > Thanks for your proactive response and appreciate the further experiment has been conducted in a short time period.
> >
> > I have checked your experiment results and the details for the feature extractor. I am happy to increase the score to 6.

---

> > > ### Author Response · Authors · 2024-11-29
> > >
> > > We would like to sincerely thank you once again for your thoughtful comments and suggestions, as well as the time you dedicated to reviewing our work.

---

### Official Review · Reviewer_2rgn · 2024-11-10

**Soundness:** 3
**Presentation:** 2
**Contribution:** 2
**Rating:** 5
**Confidence:** 3

**Summary:**

The paper presents a Fourier-based ensemble training method to counter adversarial transferability by diversifying non-robust frequency components across sub-models. Using random noise and targeted attack transformations, the method achieves improved robustness and competitive accuracy on datasets like CIFAR-10, outperforming current ensemble techniques.

**Strengths:**

- The use of Fourier transformations to manipulate frequency components for adversarial robustness is innovative and less explored in ensemble methods.
- The proposed method addresses the transferability of adversarial attacks more effectively by diversifying non-robust features across sub-models.
- Unlike simultaneous training methods, the approach’s independent training of sub-models reduces GPU memory requirements and simplifies the training pipeline.

**Weaknesses:**

- The proposed targeted-attack transformation method, while effective, introduces a layer of complexity to the training pipeline. By replacing specific non-robust frequency components in the data with adversarially targeted features, the transformation becomes computationally intensive and intricate to implement, especially when scaling to larger datasets or real-time applications. This additional complexity may limit the ease of adoption for other researchers or practitioners looking to implement this technique without substantial computational resources.
- The experiments are primarily conducted on smaller architectures, like ResNet-20, with limited exploration on larger or more complex models. Without extensive testing across a variety of architectures, it’s unclear whether the proposed Fourier-based transformations are universally effective or if they are more suitable for specific model types.

**Questions:**

Refer to the weakness.

---

> ### Author Response · Authors · 2024-11-23
> **Rebuttal**
>
> **The proposed targeted-attack transformation method, while effective, introduces a layer of complexity to the training pipeline. By replacing specific non-robust frequency components in the data with adversarially targeted features, the transformation becomes computationally intensive and intricate to implement, especially when scaling to larger datasets or real-time applications. This additional complexity may limit the ease of adoption for other researchers or practitioners looking to implement this technique without substantial computational resources.**
>
> Thanks for the concern. We agree with the reviewer that targeted-attack transformation method indeed introduce some computational cost. But this issue may not be that serious in practice. The computational cost primarily arises from the Fourier transform and obtaining adversarially targeted features.
>
> + For the Fourier transform (FFT), its computational complexity is  $nlog(n)$ (n is the image dimension), which is acceptable compared to the complexity of neural network convolutions.
>
> + For adversarially targeted features, our proposed approach is a general framework that can be incorporated with any attack transformation (so it is quite flexible in practice). Based on the transferability of adversarial examples, we can utilize a simple network architecture as pre-trained model to perform targeted attack transformation to obtain these features, and consequently the total computational cost is not increased too much. As described in Appendix E (line 894), we utilized VGG11 in our experiments.
>
> Regarding computational resources, we would like to emphasize that a highlight of our approach is that the sub-models can be trained separately—meaning they can be trained in parallel without requiring any communication between them. When computational resources are limited, the sub-models can also be trained individually, one at a time.
>
>
>
> **The experiments are primarily conducted on smaller architectures, like ResNet-20, with limited exploration on larger or more complex models. Without extensive testing across a variety of architectures, it’s unclear whether the proposed Fourier-based transformations are universally effective or if they are more suitable for specific model types.**
>
> We thank the reviewer for raising  the concern regarding the scope of model architectures used in our experiments. While the main text primarily discusses results with ResNet-20, we also evaluated our method using the WRN28-10 architecture, as detailed in the Appendix F.4. These experiments confirm the robustness and generalizability of our Fourier-based transformations across more complex models. Additionally, we evaluate a wider variety of architectures. The following table  presents the results across different model architectures, including ResNet20, ResNet50, WRN28-10, and WRN34-10, the results suggest that our method consistently enhances robustness under various attack scenarios, demonstrating its applicability across diverse architectures.
>
> | CIFAR10                | ResNet20 | ResNet50 | WRN28-10 | WRN34-10 |
> | ---------------------- | -------- | -------- | -------- | -------- |
> | clean accuracy         | 90.02    | 93.23    | 94.18    | 94.63    |
> | FGSM ($\epsilon=0.01$) | 72.24    | 76.65    | 80.64    | 81.04    |
> | FGSM ($\epsilon=0.02$) | 58.04    | 58.59    | 60.09    | 60.92    |
> | PGD ($\epsilon=0.01$)  | 48.48    | 60.23    | 64.64    | 65.38    |
> | PGD ($\epsilon=0.02$)  | 20.01    | 24.35    | 26.00    | 27.42    |
> | BIM ($\epsilon=0.01$)  | 48.57    | 60.43    | 67.36    | 68.29    |
> | BIM ($\epsilon=0.02$)  | 16.63    | 23.57    | 32.36    | 33.86    |
> | MIM ($\epsilon=0.01$)  | 51.48    | 60.81    | 64.36    | 64.71    |
> | MIN ($\epsilon=0.02$)  | 20.09    | 24.54    | 25.64    | 26.42    |
> | AA  ($\epsilon=0.01$)  | 51.56    | 60.48    | 63.45    | 64.01    |
> | AA  ($\epsilon=0.02$)  | 19.42    | 24.21    | 25.23    | 26.39    |
> | CW  ($\lambda=0.01$)   | 56.08    | 56.55    | 57.23    | 57.52    |

---

> ### Author Response · Authors · 2024-11-28
>
> Dear Reviewer 2rgn,
>
> We would like to sincerely thank you once again for your thoughtful comments and valuable suggestions on our work. As mentioned in our previous response, we have revised the manuscript based on your feedback and hope that these changes and our responses effectively address your concerns. If you have any further comments, we would be more than happy to address them.
>
> Sincerely,
>
> The authors of Paper#9133

---

> ### Author Response · Authors · 2024-12-02
>
> We have additionally conducted experiments on the larger network architecture WRN70-16, and the results are as follows.
>
>
>
> |  clean accuracy | FGSM$(\epsilon=0.01)$ | FGSM$(\epsilon=0.02)$ | PGD$(\epsilon=0.01)$ | PGD$(\epsilon=0.02)$ | BIM$(\epsilon=0.01)$ | BIM$(\epsilon=0.02)$ | MIM$(\epsilon=0.01)$ | MIM$(\epsilon=0.02)$ | AA$(\epsilon=0.01)$ | AA$(\epsilon=0.02)$ | CW$(\lambda=0.01)$ |
> | -------------- | --------------------- | --------------------- | -------------------- | -------------------- | -------------------- | -------------------- | -------------------- | -------------------- | ------------------- | ------------------- | ------------------ |
> |  95.42          | 81.63                 | 60.94                 | 67.23                | 28.31                | 69.82                | 35.18                | 66.53                | 28.31                | 66.17               | 28.39               | 59.04              |

---

### Author Response · Authors · 2024-12-02

Dear Reviewers,

With the discussion deadline approaching (in less than 24 hours), we would like to know if our response has sufficiently addressed your concerns. We greatly appreciate your thoughtful feedback and would be happy to address any additional comments you might have.

---

### Meta-Review · Area_Chair_ujVY · 2024-12-22

**Metareview:**

This paper investigates ensemble adversarial robustness against transfer attacks. Its key idea is to diversify non-robust features across different sub-models by leveraging Fourier-based data transformations and a "weakness allocation" strategy. The empirical results demonstrate a significant improvement in robustness.

Overall, the reviewers appreciate the novelty of the proposed technique and acknowledge its effectiveness as shown by the empirical results. But meanwhile, some major concerns are raised, including 1) the introduced complexity is not well analyzed; 2) the comparisons to adversarial training methods are missing; 3) additional ablations should be included, like generalization to larger model sizes and the sensitivity of frequency thresholds; and 4) the clarity and readability of some sections could be improved.

The authors addressed most of these concerns by providing additional experiments and clarifications in their rebuttal. As a result, three reviewers rated the paper a 6 (i.e., marginally above the acceptance threshold). The only negative reviewer is 2rgn, who did not engage in the discussion. By reading the corresponding rebuttal, the AC believes the concern about computational complexity and the request for additional experiments on larger models are fully addressed. Therefore, the AC recommends accepting the submission.

**Additional Comments On Reviewer Discussion:**

The reviewers' major concerns are listed in the meta-review. The authors actively provided additional experiments and further clarifications in the rebuttal and the discussion stage. The AC believes that all these major concerns have been adequately addressed and, as a result, recommends accepting the paper.

---

### Decision · Program_Chairs · 2025-01-22

Accept (Poster)